

# lcc: an R package to estimate the concordance correlation, Pearson correlation and accuracy over time

Thiago P. Oliveira[1,2,*], Rafael A. Moral[3,*], Silvio S. Zocchi[4], Clarice G.B. Demetrio[4] and John Hinde[1]

[1] School of Mathematics, Statistics and Applied Mathematics, NUI Galway, Galway, Ireland
[2] The Insight Centre for Data Analytics, NUI Galway, Galway, Ireland
[3] Department of Mathematics and Statistics, National University of Ireland, Maynooth, Maynooth, Co. Kildare, Ireland
[4] Departamento de Ciências Exatas, Luiz de Queiroz College of Agriculture - USP, Piracicaba, São Paulo, Brazil
* These authors contributed equally to this work.

## ABSTRACT

**Background and Objective:** Observational studies and experiments in medicine, pharmacology and agronomy are often concerned with assessing whether different methods/raters produce similar values over the time when measuring a quantitative variable. This article aims to describe the statistical package lcc, for are, that can be used to estimate the extent of agreement between two (or more) methods over the time, and illustrate the developed methodology using three real examples.
**Methods:** The longitudinal concordance correlation, longitudinal Pearson correlation, and longitudinal accuracy functions can be estimated based on fixed effects and variance components of the mixed-effects regression model. Inference is made through bootstrap confidence intervals and diagnostic can be done via plots, and statistical tests.
**Results:** The main features of the package are estimation and inference about the extent of agreement using numerical and graphical summaries. Moreover, our approach accommodates both balanced and unbalanced experimental designs or observational studies, and allows for different within-group error structures, while allowing for the inclusion of covariates in the linear predictor to control systematic variations in the response. All examples show that our methodology is flexible and can be applied to many different data types.
**Conclusions:** The `lcc` package, available on the CRAN repository, proved to be a useful tool to describe the agreement between two or more methods over time, allowing the detection of changes in the extent of agreement. The inclusion of different structures for the variance-covariance matrices of random effects and residuals makes the package flexible for working with different types of databases.

Corresponding author
Thiago P. Oliveira,
thiago.paula.oliveira@usp.br

## INTRODUCTION

Agreement indices are generally used when the same experimental unit is measured by at least two methods or observers (*King et al., 2007*). Measurements of agreement between raters or methods can be used in any field to explore their interchangeability considering a certain degree of agreement between the measurements they provide (*Barnhart & Williamson, 2001*; *Chen & Barnhart, 2013*). In biomedical sciences it is often necessary to study the reproducibility of continuous measurements made using specific diagnostic tools or methods, and that measurements can be taken over the time on the subjects of interest, such as in the studies of *Pandit, Chair & Schuller (2019)*; *Shinar et al. (2019)* and *Loecher et al. (2019)*.

The concordance correlation coefficient (CCC) introduced by *Lin (1989)* is a statistic commonly used to measure the agreement between methods when the response is continuous. Let $Y_1$ and $Y_2$ be two random variables with a joint normal distribution

$$\begin{bmatrix} Y_1 \\ Y_2 \end{bmatrix} \sim N_2 \left( \begin{bmatrix} \mu_1 \\ \mu_2 \end{bmatrix}, \ \Sigma = \begin{bmatrix} \sigma_1^2 & \sigma_{12} \\ \sigma_{12} & \sigma_2^2 \end{bmatrix} \right).$$

Here the expected value of the squared difference between $Y_1$ and $Y_2$ can be used as an agreement value. However, it ranges from 0 (perfect agreement) to infinity, which makes its interpretation difficult. *Lin (1989)* proposed standardizing this agreement index so that its values lie between −1 and +1:

$$\rho_{\text{CCC}} = 1 - \frac{\text{E}\left[ (Y_1 - Y_2)^2 \right]}{\sigma_1^2 + \sigma_2^2 + (\mu_1 - \mu_2)^2} = \frac{2\sigma_{12}}{\sigma_1^2 + \sigma_2^2 + (\mu_1 - \mu_2)^2} = \rho C_b$$

where $\mu_1 = \text{E}(Y_1)$, $\mu_2 = \text{E}(Y_2)$, $\sigma_1^2 = \text{Var}(Y_1)$, $\sigma_2^2 = \text{Var}(Y_2)$ and $\sigma_{12} = \text{Cov}(Y_1, Y_2)$. This coefficient takes the value −1 when there is perfect disagreement, zero when there is no agreement, and +1 when there is perfect agreement. Moreover, $\rho$, the Pearson correlation coefficient ($|\rho| \leq 1$), measures how far each observation deviated from the best-fit line (a precision measure) and $C_b$, the accuracy ($0 < C_b \leq 1$), measures how far the best-fit line deviates from the 45° line through the origin, defined as $C_b = 2(v + v^{-1} + u^2)^{-1}$, where $v = \sigma_1^2/\sigma_2^2$ is a scale shift and $u = (\mu_1 - \mu_2)/\sqrt{\sigma_1 \sigma_2}$ is a location shift relative to the scale (*Lin, 1989*). Note that $C_b = 1$ indicates no deviation from the 45° line. In an attempt to improve the inferential ability, *Liao (2003)* extended the concordance correlation coefficient by using two random paired measurements to the identity line.

When pairs of samples $(Y_{i1k}, Y_{i2k})$, for $i = 1, 2, \ldots, N$ subjects and $k = 1, 2, \ldots K$ repeated measures, corresponding to observations on the same subject or experimental unit over time, the use of generalized multivariate analysis of variance to compute a weighted version of the CCC for repeated measurements is recommended (*Chinchilli et al., 1996*). Moreover, this coefficient has also been expanded to assess the agreement between more than two methods (*King & Chinchilli, 2001*).

When it is necessary to add extra variability sources due to within-subject measurements and/or other covariates in the model, the CCC can be estimated through the variance components (VC) of a mixed-effects model (*Carrasco, King & Chinchilli, 2009*). The advantages of the mixed-effects models are that they give a general approach to analyse repeated measures and unbalanced data; they allow for the inclusion of different variance-covariance structures for both random effects and sampling errors. The restricted maximum likelihood (REML) approach can be used to obtain unbiased estimates of the VC.

Nevertheless, sometimes the researcher is not interested in reducing the CCC for repeated measurements to a single value, as proposed by *Carrasco, King & Chinchilli (2009)* and *Carrasco et al. (2013)*, but in describing the extent of agreement between methods over time, as discussed by *Liao (2005)* in a non-parametric case. However, in the parametric case, we can consider a linear or non-linear function of the time and/or covariates in the model to describe the response variable, as proposed by *Rathnayake & Choudhary (2017)* and *Oliveira, Hinde & Zocchi (2018)*. Here, we present the implementation of this methodology as an R (*R Core Team, 2019*) package `lcc` (*Oliveira et al., 2019*), which provides functions for estimating the longitudinal concordance correlation (LCC) between methods based on variance components and fixed effects using polynomial mixed-effects models. It also computes estimates for the longitudinal Pearson correlation (LPC), which measures the precision, and the longitudinal bias correction factor (LA), which provides an accuracy measure.

The `lcc()` function gives fitted values and non-parametric bootstrap confidence intervals for the LCC, LPC and LA statistics. Moreover, they can be estimated using different structures for the variance-covariance matrices of the random effects and different variance functions to model heteroskedasticity of within-group errors, with the option of using time as a variance covariate.

The remainder of the article is organized as follows: Section 1 introduces the theoretical definition of the LCC. Section 2 introduces the `lcc()` function input and output, describing in detail the various options as well the `summary()` and other generic methods. Section 3 briefly discusses model specification, which is illustrated more extensively in Section 4 using three real data examples. The first and third shows an application in biomedical science, while the second from food science was the motivation for the development of the methodology and software and nicely shows the utility of the approach. Section 5 provides a discussion about the lcc package, and the importance of LPC and LA. Finally, Section 6 presents some final remarks about the `lcc` package.

## MODELS AND COMPUTATIONAL METHODS

Suppose a researcher is interested in investigating the extent of agreement between two or more methods, indexed as $j = 1, 2, \ldots, J$. Let $N$ be the number of subjects in the experiment or observational study, indexed as $i = 1, 2, \ldots, N$, and suppose that each subject

is observed $n_i$ times (visits) with associated nuisance factors and/or covariates, these could include, for example, the effect of block or group. Let $y_{ijk}$ be a realization of a random variable $Y_{ijk}$ measured on the $i$-th subject by the $j$-th method at time $t_k$, $k = 1, 2, \ldots, n_i$, with additional subject level (nuisance) covariates $\boldsymbol{x}_i$. Here $t_k$ assumes values of the time covariate $t \in \mathcal{T}$, where $\mathcal{T}$ denotes the set of measurement times. Hence, the linear mixed-effects model including a polynomial function of time per method, random effects of subject, as well random effects for as subject/time interactions, is given by

$$Y_{ijk} = \boldsymbol{\gamma}^T \boldsymbol{x}_i + \sum_{h=0}^{p} \beta_{hj} t_{ik}^h + \sum_{h=0}^{q} b_{hi} t_{ik}^h + \varepsilon_{ijk}, \tag{1}$$
$$\text{with } \mathbf{b}_i \sim \mathrm{MVN}(\mathbf{0}, \mathbf{G}) \text{ and } \varepsilon_i \sim \mathrm{MVN}(\mathbf{0}, \mathbf{R}_i)$$

where $h = 1, 2, \ldots, q, q + 1, \ldots, p$ is an index identifying the degree of the polynomial, with $q \leq p$; $Y_{ijk}$ is the response measured on the $i$-th subject by the $j$-th method at time $t_{ik}$; $t_{ik}$ represents the time (seconds, minutes, days, etc) at which the $i$-th individual was observed; $\boldsymbol{\gamma}$ is a vector of fixed effect parameters for the subject level covariates; $\boldsymbol{\beta}_j = \left[ \beta_{0j}, \beta_{1j}, \ldots, \beta_{pj} \right]^T$ is a $(p + 1)$-dimensional vector of fixed effects for the $j$-th method; $\mathbf{b}_i = \left[ b_{0i}, b_{1i}, , b_{qi} \right]^T$ is a $(q + 1)$-dimensional vector of random effects with mean vector $\mathbf{0}$ and covariance matrix $\mathbf{G}$; $\boldsymbol{\varepsilon}_i$ is a $(J \times n_i)$-dimensional error vector assumed to be independent for different $i$ and independent of the random effects, with independent entries over $j$ and $k$, with mean vector $\mathbf{0}$ and diagonal variance matrix $\boldsymbol{R}_i$.

Under model (1), the longitudinal concordance correlation (LCC) function between methods $j$ and $j'$, $j \neq j'$, is given by

$$\rho_{jj'}(t_k) = \frac{\mathbf{t}_k \mathbf{G} \mathbf{t}_k^T}{\mathbf{t}_k \mathbf{G} \mathbf{t}_k^T + \frac{1}{2} \left\{ \sigma_\varepsilon^2 \left[ g(t_k, \boldsymbol{\delta}_j) + g(t_k, \boldsymbol{\delta}_{j'}) \right] + S_{jj'}^2(t_k) \right\}} = \rho_{jj'}^{(p)}(t_k) C_{jj'}(t_k) \tag{2}$$

where $S_{jj'}(t_k) = \mathbf{t_k} \left( \boldsymbol{\beta}_j - \boldsymbol{\beta}_{j'} \right)$ is the systematic difference between methods $j$ and $j'$; $\mathbf{t}_k^T = \left( t_k^0, t_k^1, \ldots, t_k^q \right)^T$; $g(\cdot)$ is a variance function assumed continuous in $\boldsymbol{\delta}$; $\boldsymbol{\delta}_j$ is a vector of variance parameters for observations measured by $j$-th method or observer. We have that $\rho_{jj'}^{(p)}(t_k)$ is the longitudinal Pearson correlation (LPC) that measures how far each observation deviated from the best-fit line at a fixed time $t_k = t$, given by

$$\rho_{jj'}^{(p)}(t_k) = \frac{\mathbf{t}_k \mathbf{G} \mathbf{t}_k^T}{\sqrt{\left[ \mathbf{t}_k \mathbf{G} \mathbf{t}_k^T + \sigma_\varepsilon^2 g(t_k, \boldsymbol{\delta}_j) \right] \left[ \mathbf{t}_k \mathbf{G} \mathbf{t}_k^T + \sigma_\varepsilon^2 g(t_k, \boldsymbol{\delta}_{j'}) \right]}}.$$

$C_{jj'}(t_k)$, the longitudinal accuracy (LA), measures how far the best-fit line deviates from the $45°$ line at a fixed time $t_k = t$, given by

$$C_{jj'}(t_k) = \frac{2}{v_{jj'}(t_k) + \left[ v_{jj'}(t_k) \right]^{-1} + u_{jj'}^2(t_k)},$$

where

$$v_{jj'}(t_k) = \sqrt{\frac{\mathrm{Var}(Y_{ijkl})}{\mathrm{Var}(Y_{ij'kl})}} = \sqrt{\frac{\mathbf{t}_k \mathbf{G} \mathbf{t}_k^T + \sigma_\varepsilon^2 g(t_k, \boldsymbol{\delta}_j)}{\mathbf{t}_k \mathbf{G} \mathbf{t}_k^T + \sigma_\varepsilon^2 g(t_k, \boldsymbol{\delta}_{j'})}}$$

denotes the scale shift at time $t_k = t$, and

$$u_{jj'}(t_k) = \frac{\mathrm{E}(Y_{ijkl}) - \mathrm{E}(Y_{ij'kl})}{\left[\mathrm{Var}(Y_{ijkl})\mathrm{Var}(Y_{ij'kl})\right]^{\frac{1}{4}}}$$

$$= \frac{\mathbf{t}_k\left(\boldsymbol{\beta}_j - \boldsymbol{\beta}_{j'}\right)}{\left\{\left[\mathbf{t}_k \mathbf{G} \mathbf{t}_k^T + \sigma_\varepsilon^2 g(t_k, \boldsymbol{\delta}_j)\right]\left[\mathbf{t}_k \mathbf{G} \mathbf{t}_k^T + \sigma_\varepsilon^2 g(t_k, \boldsymbol{\delta}_{j'})\right]\right\}^{\frac{1}{4}}}$$

denotes the location shift at time $t_k$ relative to the scale (*Lin, 1989*; *Oliveira, Hinde & Zocchi, 2018*). Consequently, when $\mathrm{Var}(Y_{ijkl}) = \mathrm{Var}(Y_{ij'kl})$ and $\mathrm{E}(Y_{ijkl}) = \mathrm{E}(Y_{ij'kl})$ then $C_{jj'}(t_k) = 1$ and there is no deviation from the 45° line.

## Estimation and inference

Point estimation and statistical inference for the LCC $\left(\rho_{jj'}(t_k)\right)$ has been proposed by *Oliveira, Hinde & Zocchi (2018)*. It is estimated by replacing $\boldsymbol{\beta}$ and the variance components by their respective REML estimates:

$$\widehat{\rho}_{jj'}(t_k) = \frac{\mathbf{t}_k \widehat{\mathbf{G}} \mathbf{t}_k^T}{\mathbf{t}_k \widehat{\mathbf{G}} \mathbf{t}_k^T + \frac{1}{2}\left\{\widehat{\sigma}_\varepsilon^2\left[\widehat{g}\left(t_k, \widehat{\boldsymbol{\delta}}_j\right) + \widehat{g}\left(t_k, \widehat{\boldsymbol{\delta}}_{j'}\right)\right] + \widehat{S}_{jj'}^2(t_k)\right\}}.$$

Since the variance components are estimated using the REML approach, their estimates are asymptotically normally distributed and the bias is smaller when compared to the maximum likelihood (ML) approach. Moreover, *Oliveira, Hinde & Zocchi (2018)* showed a satisfactory performance of the LCC even in settings with severe imbalance and only a small number of subjects ($N = 20$).

A confidence interval (CI) for $\rho_{jj'}(t_k)$ can be constructed using a nonparametric bootstrap based on $M$ (e.g., 5,000) bootstrap samples with either the percentile method (recommended for $N \le 30$) or, otherwise, a normal approximation confidence interval, as described by *Oliveira, Hinde & Zocchi (2018)*.

When we use a normal approximation for the CI, the Fisher Z-transformation given by

$$\rho_{j,j'}^*(t_k) = \frac{1}{2}\ln\left[\frac{1 + \rho_{j,j'}(t_k)}{1 - \rho_{j,j'}(t_k)}\right]$$

should be used with the normal approximation made to the empirical distribution of $\rho_{j,j'}^*(t_k)$ (*Lin, 1989*). Consequently, the confidence limits can be estimated using the bootstrap estimator of $\rho_{j,j'}^*(t_k)$ for a fixed time $t_k = t$ given by

$$\widehat{\rho}_{j,j'}^*(t_k = t) = \frac{1}{2M}\sum_{m=1}^{M} \ln\left[\frac{1 + \widehat{\rho}_{j,j'}^{(m)}(t)}{1 - \widehat{\rho}_{j,j'}^{(m)}(t)}\right], \quad m = 1, 2, \ldots, M,$$

where $\left\{\widehat{\rho}_{j,j'}^{(m)}\right\}$ are the estimates from the $M$ bootstrap samples. The standard deviation of the bootstrap distribution of $\widehat{\rho}_{j,j'}^{*}(t_k)$ for a fixed time $t_k = t$ given by

$$\widehat{SE}_{j,j'}^{*}(t_k = t) = \sqrt{\frac{1}{M-1}\sum_{m=1}^{M}\left[\frac{1}{2}\ln\left(\frac{1+\widehat{\rho}_{j,j'}^{(m)}(t)}{1-\widehat{\rho}_{j,j'}^{(m)}(t)}\right) - \widehat{\rho}_{j,j'}^{*}(t)\right]^2}.$$

Thus, an approximate bootstrap confidence interval of level $(1-\alpha)$ for $\rho_{j,j'}$ is $[LB, UB]$, where

$$LB = \frac{\exp\left\{2\left[\widehat{\rho}_{j,j'}^{*}(t_k = t) - z_{(1-\frac{\alpha}{2})}\widehat{SE}_{j,j'}^{*}(t_k = t)\right]\right\} - 1}{\exp\left\{2\left[\widehat{\rho}_{j,j'}^{*}(t_k = t) - z_{(1-\frac{\alpha}{2})}\widehat{SE}_{j,j'}^{*}(t_k = t)\right]\right\} + 1}$$

and

$$UB = \frac{\exp\left\{2\left[\widehat{\rho}_{j,j'}^{*}(t_k = t) - z_{\frac{\alpha}{2}}\widehat{SE}_{j,j'}^{*}(t_k = t)\right]\right\} - 1}{\exp\left\{2\left[\widehat{\rho}_{j,j'}^{*}(t_k = t) - z_{\frac{\alpha}{2}}\widehat{SE}_{j,j'}^{*}(t_k = t)\right]\right\} + 1},$$

where $z_{\frac{\alpha}{2}}$ and $z_{(1-\frac{\alpha}{2})}$ denote the $\frac{\alpha}{2}$ and $(1-\frac{\alpha}{2})$ percentiles of the standard normal distribution.

On the other hand, the CI based on the percentile method uses the percentiles of the bootstrap distribution of $\widehat{\rho}_{j,j'}(t_k = t)$ directly and is given by

$$\left(\widehat{\rho}_{j,j'_{(\alpha/2)}}(t_k = t), \widehat{\rho}_{(j,j')_{(1-\alpha/2)}}(t_k = t)\right) \approx \left(\widehat{\rho}_{j,j'_{(\alpha/2)}}^{(m)}(t_k = t), \widehat{\rho}_{(j,j')_{(1-\alpha/2)}}^{(m)}(t_k = t)\right)$$

where $\widehat{\rho}_{(j,j')_{(\alpha/2)}}^{(m)}(t_k = t)$ and $\widehat{\rho}_{(j,j')_{(1-\alpha/2)}}^{(m)}(t_k = t)$ are the $(100 \times \frac{\alpha}{2})$−th and $(100 \times 1 - \frac{\alpha}{2})$−th empirical percentiles of the $\widehat{\rho}_{j,j'}^{(m)}(t_k = t)$ values, $m = 1, 2, \ldots, M$. If the bootstrap distribution of $\rho_{j,j'}^{*}(t_k = t)$ is approximately normal, then both proposed methods will give very similar confidence intervals as $N$ increases.

Inference for $C_{jj'}(t_k)$ can be performed in a similar way as to that presented for the LCC. Since $C_{(j,j')_{(1-\alpha/2)}}(t_k = t)$ belongs to the interval $[0, 1]$, we suggest the use the arc-sine transformation

$$C_{(j,j')_{(1-\alpha/2)}}^{*}(t_k = t) = \sin^{-1}\sqrt{C_{jj'}(t_k)}$$

instead of the Fisher Z-transformation, nor logistic transformation (used by *Oliveira, Hinde & Zocchi (2018)*) to approximate the distribution of $C_{(j,j')_{(1-\alpha/2)}}(t_k = t)$ by a normal distribution. Thus, the confidence limits can be estimated using the bootstrap estimator of $C_{j,j'}^{*}(t_k)$ for a fixed time $t_k = t$ given by

$$\widehat{C}_{j,j'}^{*}(t_k = t) = \frac{1}{M}\sum_{m=1}^{M}\sin^{-1}\sqrt{\widehat{C}_{j,j'}^{(m)}(t)}, \quad m = 1, 2, \ldots, M,$$

and standard deviation of the bootstrap distribution of $\widehat{C}^*_{j,j'}(t_k)$ for a fixed time $t_k = t$ is given by

$$\widehat{SE}^*_{C_{j,j'}}(t_k = t) = \sqrt{\frac{1}{M-1} \sum_{m=1}^{M} \left[ \sin^{-1} \sqrt{\widehat{C}^{(m)}_{j,j'}(t)} - \widehat{C}^*_{j,j'}(t) \right]^2}$$

Therefore, an approximate bootstrap confidence interval of level $(1 - \alpha)$ for $\widehat{C}_{j,j'}$ is $[LB_C, UB_C]$, where

$$LB_C = sign\left[ \widehat{C}^*_{j,j'}(t_k = t) - z_{(1-\frac{\alpha}{2})} \widehat{SE}^*_{C_{j,j'}}(t_k = t) \right] \left\{ \sin\left[ \widehat{C}^*_{j,j'}(t_k = t) - z_{(1-\frac{\alpha}{2})} \widehat{SE}^*_{C_{j,j'}}(t_k = t) \right] \right\}^2$$

and

$$UB_C = sign\left[ \widehat{C}^*_{j,j'}(t_k = t) - z_{\frac{\alpha}{2}} \widehat{SE}^*_{C_{j,j'}}(t_k = t) \right] \left\{ \sin\left[ \widehat{C}^*_{j,j'}(t_k = t) - z_{\frac{\alpha}{2}} \widehat{SE}^*_{C_{j,j'}}(t_k = t) \right] \right\}^2,$$

where $z_{\frac{\alpha}{2}}$ and $z_{(1-\frac{\alpha}{2})}$ denote the $(\frac{\alpha}{2})$ and $(1 - \frac{\alpha}{2})$ quantiles of the standard normal distribution. Bootstrap percentile intervals are calculated in the obvious way from the bootstrap values $\widehat{C}^{(m)}_{j,j'}(t)$, $m = 1, 2, \ldots, M$.

## OVERVIEW OF THE PACKAGE `lcc` AND R SYNTAX

This section provides some details on the implementation of the function lcc and explains its technical arguments, whose default settings were carefully chosen. The package is freely available for download from the CRAN website https://CRAN.R-project.org/package=lcc, and installation can be performed using

```
R> install.packages("lcc")
R> library(lcc)
```

The `lcc` package has 21 arguments that are briefly summarised in Table 1.

We present a more detailed description of some arguments below:

1. `data`: must be a data frame containing the following variables: response, subject identification, method, and time;

2. `method`: name of the method variable in the dataset. The `lcc` package recognizes the first level of the variable associated with this argument as the gold-standard method, and then compares it with all other levels;

3. `qr`: when we specify `qr = 0` a random intercept is included in the polynomial model while `qr = 1` specifies random intercepts and slopes. If `qr = qf = q`, with $q \geq 1$, all polynomial terms are specified to have random effects at the individual level.

4. `time_lcc`: a named list with values for arguments `time`, `from`, `to`, and `n` used in the `time_lcc()` function to generate a regular sequence merged with specific or experimental time values of the time variable used for LCC, LPC and LA predictions. Argument `time` is a vector of specific or experimental time values of a given length, where the experimental time values are used as default; `from` and `to` are used to define, respectively, the starting and end values of the time variable, and `n` is used to define the desired length of the sequence. We recommend a grid $\mathbf{t} = (t_1, t_2, \ldots, t_{n^*})^T$ of $n^*$ points in $\mathcal{T}$ to construct the

agreement curve and confidence intervals. In practice, $n^*$ between 30 and 50 is generally adequate. Example:

```
R> Time <- seq(0,20,1)
R> str(tk <- time_lcc(time=Time, from=min(Time), to=max(Time),
+ n=30))
num [1:49] 0 0.69 1 1.38 2 ...
```

5. `pdmat`: the `lcc` package provides six standard classes of positive-definite matrix structures that can be included in the model to estimate the LCC, LPC and LA statistics. Available standard classes are `pdSymm`, `pdLogChol`, `pdDiag`, `pdIdent`, `pdCompSymm` and `pdNatural`. More information about these classes are available in *Pinheiro & Bates (2000)*.

6. `var.class`: a class of variance functions that are used to model the variance structure of within-group errors using covariates (*Pinheiro & Bates, 2000*). We generalize this class as

$$\text{Var}\left(\varepsilon_{ijk}\right) = \sigma_\varepsilon^2 g(t_k, \boldsymbol{\delta}), \tag{3}$$

where $g(\cdot)$ is the variance function assumed continuous in $\boldsymbol{\delta}$; $t_k$ is the time covariate and $\boldsymbol{\delta}$ is a vector of variance parameters. The `lcc` package provides two different standard variance functions classes that are included in the `nlme` library (*Pinheiro et al., 2017*).

The first one is the `varIdent` class that represent a variance model with different variances for each level of a stratification variable $s$, $s = 1, 2, \ldots, S$, given by

$$\text{Var}\left(\varepsilon_{ijk}\right) = \sigma_\varepsilon^2 \delta_{s_{ijk}}^2.$$

As we have $S + 1$ parameters to represent $S$ variances, we need to add the restriction $\boldsymbol{\delta}_1 = 1$, and consequently $\boldsymbol{\delta}'_{s^*} = \boldsymbol{\delta}_{s^*}/\boldsymbol{\delta}_1$, $s^* = 2, 3, \ldots, S$ and $\boldsymbol{\delta}'_{s^*} > 0$. Here each level of method/observer or time represents a stratum of a homogeneous subgroup.

The second variance function is an exponential function of the variance covariate, the `varExp` class, represented as

$$\text{Var}\left(\varepsilon_{ijk}\right) = \sigma_\varepsilon^2 \exp\left(2\delta_{s_{ijk}} t_k\right)$$

where $\delta_{\text{sijk}}$ is unrestricted, so the variance model (4) allows $\text{Var}\left(\varepsilon_{ijk}\right)$ to increase or decrease over time.

7. `weights.form`: a `varFunc` class object, representing a constructor to the `form` argument in the `nlme` library. The `weights.form` argument is based on a one-sided formula specifying a variance covariate and, optionally, a grouping factor for the variance parameters. Moreover, this argument must be specified only when `var.class` is specified as well.

The first class `varIdent` represents a variance model with different variances for each level of the grouping factor and has two options of `weights.form` in the `lcc` package:

(a) "`method`": specifies a variance model with different variances for each level of factor method/observer and is given by

$$\text{Var}\left(\varepsilon_{ijk}\right) = \sigma_\varepsilon^2 \delta_{\text{method}_j}^2, \quad j = 1, 2, \ldots, J,$$

**Table 1 Input arguments for LCC package.**

| Argument | Type | Description | Default | Required |
|---|---|---|---|---|
| data | data. frame | Specifies the input dataset | | Yes |
| resp | Character string | Name of the response variable | | Yes |
| subject | Character string | Name of the subject variable | | Yes |
| method | Character string | Name of the method variable | | Yes |
| time | Character string | Name of the time variable | | Yes |
| interaction | Logical | An option to estimate the interaction effects between method and time. If `TRUE` the interaction effects are estimated. If `FALSE` only the main effects of time and method are estimated | TRUE | No |
| qf | Numeric | An integer specifying the degree of the polynomial time trends, usually 1, 2 or 3 (0 is not allowed). | 1 | No |
| qr | Numeric | An integer specifying terms having random effects to account for subject-to-subject variation, such that $qr \leq qf$, and qr=0 means there is just a random intercept. | 0 | No |
| covar | Character vector | Names of the covariates (factors and/or variables) to include in the model as fixed effects, for example, block, group, etc. | NULL | No |
| gs | Character string | Name of method level which represents the gold-standard. | first level | No |
| pdmat | Function | Standard classes of positive-definite matrix structures available in the `nlme` package. | pdSymm | No |
| var.class | Function | Standard classes of variance function structures used to model the variance structure of within-group errors using covariates. | NULL | No |
| weights.form | Formula | An one-sided formula specifying a variance covariate and, optionally, a grouping factor for the variance parameters in the `var.class`. If `var.class = varIdent`, the form "method", (or ~ 1 \| method), or "time.ident" (~ 1 \| time), must be used. If `var.class = varExp`, the form "time" ( ~ time), or "both" (~ time \| method), must be used. | NULL | No[1] |
| time_lcc | List | Regular sequence for time variable merged with specific or experimental time values used for LCC, LPC, and LA predictions. | NULL | No |
| ci | Logical | An optional non-parametric boostrap confidence interval for the LCC, LPC and LA statistics. If `TRUE` confidence intervals are calculated and printed in the output. | FALSE | No |
| percentileMet | Logical | An optional method for calculating the non-parametric bootstrap intervals. If `FALSE` the normal approximation method is used. If `TRUE` the percentile method is used. | FALSE | No[2] |
| alpha | Numeric | Confidence level for the CI. | 0.05 | No[2] |
| nboot | Numeric | An integer specifying the number of bootstrap samples. | 5,000 | No[2] |
| show.warnings | Logical | An optional argument that shows the number of convergence errors in the bootstrap samples. If `TRUE` shows in which bootstrap samples the errors occurred. If `FALSE` shows the total number of convergence errors. | FALSE | No |
| components | Logical | An option to estimate the LPC and LA statistics. If `TRUE` the estimates and confidence intervals for LPC and LA are printed in the output. If `FALSE` provides estimates and confidence intervals only for the LCC statistic. | FALSE | No |
| REML | Logical | The estimation method. If `TRUE` the model is fit by maximizing the restricted log-likelihood. If `FALSE` full maximum likelihood is used. | TRUE | No |
| lme.control | List | A list of control values passed to the estimation algorithm to replace the default values of the function `lmeControl` available in the `nlme` package. | empty list | No |
| numCore | Integer | Number of cores used in parallel during bootstrapping computation | 1 | No |

Notes:
[1] Required when `var.class` is specified.
[2] It can only be specified when `ci = TRUE`.

where $g(\text{method}_j, \delta_j) = \delta^2_{\text{method}_j}$ is the variance function, and $\delta_{\text{methodj}}$ is the variance parameter for observations measured by the $j$th method. The `form` argument in the `varFunc` is `form = ~ 1|method`;

(b) "`time.ident`": specifies a variance model with different variances for each level of stratification in the time variable and is given by

$$\text{Var}(\varepsilon_{ijk}) = \sigma^2_\varepsilon \delta^2_{t_k}, \quad k = 1, 2, \ldots, K,$$

where $g(t_k, \delta_k) = \delta^2_{t_k}$ is the variance function, and $\delta_{tk}$ is the variance parameter for observations measured at time $t_k = t$, with $t \in [t_0, t_K]$ and $t_0 \geq 0$. The `form` argument in the `varFunc` class is `form = ~ 1|time`.

The class `varExp` represents a variance model whose variance function $g(.)$ is an exponential function of the variance covariate. This class has also two options of `weights.form` in the `lcc` package:

(a) "`time`": specifies a variance model given by

$$\text{Var}(\varepsilon_{ijk}) = \sigma^2_\varepsilon \exp(2\delta t_k),$$

where the variance function $g(t_k, \delta) = \exp(2\delta t_k)$ is an exponential function of the time $t_k = t$; and $\delta$ is the variance parameter. The `form` argument in the `varFunc` class is `form = ~ time`;

(b) "`both`": specify a variance model for each level of the factor method given by

$$\text{Var}(\varepsilon_{ijk}) = \sigma^2_\varepsilon \exp(2\delta_{\text{method}_j} t_k), \quad j = 1, 2, \ldots, J,$$

where the variance function $g(t_k, \text{method}_j, \delta) = \exp(2\delta_{\text{method}_j} t_k)$ is an exponential function of the time $t_k = t$ for each level of method; and $\delta_{\text{methodj}}$ is the variance parameter for the $j$th level of method. The form argument in the `varFunc` class is `form = ~ time| method`;

The `lcc` package uses the REML method as default because it is less biased, less sensitive to outliers, and deals more effectively with high correlations when compared to standard ML estimation (*Harville, 1977*; *Giesbrecht & Burns, 1985*). However, we offer the user the possibility to change the estimation method to ML because this approach should be used when comparing models with nested fixed effects but with the same random effects structure. Furthermore, the package depends on the `nlme` (*Pinheiro et al., 2017*) and `ggplot2` (*Wickham, 2009*) packages, and imports some functions from packages `gdata` (*Warnes et al., 2017*), `gridExtra` (*Auguie & Antonov, 2017*) and `hnp` (*Moral, Hinde & Demétrio, 2017*).

## Generic functions and outputs

A typical call of the `lcc` function is similar to a call to `lme` as the LCC estimation is based on a mixed-effects regression model. Several variations in the specifications of linear mixed-effects models to estimate the LCC are possible, and we can query the fitted `lcc`

**Table 2 Generic functions for use with objects of class lcc.**

| Function | Description |
|---|---|
| print() | A simple printed display |
| summary() | Returns an object of class summary.lcc containing the relevant summary statistics (which has a print() method). If type = "lcc" it provides information about $\rho_{jj'}(t_k)$, and if components = TRUE in the lcc() function, also provides information about $\rho_{jj'}^{(p)}(t_k)$, and $C_{jj'}(t_k)$. If type = "model" it provides additional information about the linear mixed-effects fit. The default is type = "model". |
| anova() | Summarise and compare likelihoods of fitted models from lcc objects |
| coef() | The fixed effects estimated and corresponding random effects estimates are obtained at subject levels less or equal to $N$. The resulting estimates are returned as a data frame, with rows corresponding to subject levels and columns as coefficients. |
| fitted() | Fitted values for $\widehat{\rho}_{jj'}(t_k)$, $\widehat{\rho}_{jj'}^{(p)}(t_k)$, or $\widehat{C}_{jj'}(t_k)$. The output depends on the argument type, where type = "lcc" (the default), type = "lpc", or type = "la" gives output for $\widehat{\rho}_{jj'}(t_k)$, $\widehat{\rho}_{jj'}^{(p)}(t_k)$, or $\widehat{C}_{jj'}(t_k)$, respectively. |
| getVarCov() | Returns the variance components estimates. |
| residuals() | Extract residuals (response, Pearson, and normalized), defaulting to Pearson. residuals |
| ranef() | Extract the estimated random effects. |
| vcov() | Returns the variance-covariance matrix of the fixed effects. |
| AIC() | Compute the Akaike criterion |
| BIC() | Compute the Bayesian criterion |
| logLik() | Extract the log-likelihood |
| plot() | A series of six built-in diagnostic plots to evaluate the assumptions underlying the linear mixed-effects regression model. Comprises: a plot of conditional residuals against fitted values; plot of conditional residuals over time; box-plot of residuals given subject; observed against fitted values; normal Q-Q plot with simulation envelopes for the conditional errors; and normal Q-Q plot with simulation envelopes for the random effects are provided. |

object through different generic functions. Table 2 gives details of a set of S3 generic extractor functions for objects of class lcc.

The output of the summary() function includes the values of Akaike Information Criterion (AIC) (*Akaike, 1974*), the Bayesian Information Criterion (BIC) (*Schwarz, 1978*), log-likelihood value, and a goodness of fit measurement gof, which is calculated using the concordance correlation coefficient (*Lin, 1989*) between fitted values extracted from the mixed-effects model and observed values. This measure can be used, with care, to describe the overall agreement between observed and fitted values, where a value equal to −1 represents a perfect disagreement between them, zero represents no agreement, and +1 perfect agreement. Clearly, a high model performance is related with a high positive value of gof (generally between 0.8 and 1).

The fitted curves of LCC, LPC, or LA values versus the time covariate, as well as their bootstrap confidence intervals, can be visualised through the lccPlot() function, which is specified as lccPlot(obj, type, control), where obj is an object of class lcc; type specifies required output that could be type="lcc" for the LCC, the default, type="lpc" for the LPC, or type="la" for the LA statistics; and control is a list of control values or character strings returned by the plotControl() function used to modify the plot structure. This function uses the ggplot2 package internally to build the final plot, where predicted values are joined by lines, sampled observations are represented by circles, and confidence intervals by a ribbon (grey as default) defined by its lower and upper bounds.

## SPECIFYING MODELS IN THE `lcc()` FUNCTION

In the `lcc` package, to describe the LCC we need to specify the subject, response, method and time variables, a polynomial mixed-effect model, and the data. These arguments are specified through an easy-to-use syntax. Consider a first degree polynomial model with random intercepts for a continuous dependent variable $y$ observed on $N$ subjects ($i = 1, 2, \ldots, N$) using $J$ methods at times $t_k$ ($k = 1, 2, \ldots, n_i$). Such model can be written as

$$Y_{ijk} = \beta_{0j} + b_{0i} + \beta_{1j}t_k + \varepsilon_{ijk}, \text{ with } b_{0i} \sim N(0, \sigma_{b_0}^2) \quad \text{and} \quad \varepsilon_{ijk} \sim N(0, \sigma_\varepsilon^2)$$

Thus, the LCC based on fixed effects and variance components at time $t_k$ is given by

$$\rho_{jj'}(t_k) = \frac{\sigma_{b_0}^2}{\sigma_{b_0}^2 + \sigma_\varepsilon^2 + \frac{1}{2}[\beta_{01} - \beta_{02} + (\beta_{11} - \beta_{12})t_k]^2}$$

and the syntax to specify this model in the `lcc()` function is

```
R> library(lcc)
R> data(simulated_hue_block)
R> m1 <- lcc(data = simulated_hue_block, subject = "Fruit",
+            resp = "Hue", method = "Method", time = "Time",
+            qf = 1, qr = 0)
```

where `qf = 1` represents the polynomial degree for the fixed effects, and `qr = 0` specifies a random intercepts model. Here, the names of the columns in the dataframe `data` are supplied as strings to the arguments of the `lcc()` function.

Suppose now that the experimental design in the previous example was a randomized complete block design. Then, the fixed effect of blocks can be included in that model by specifying the `covar` argument, that is,

```
R> m2 <- update(m1, covar ="Block")
```

If we suppose different variances for each level of the method factor, the corresponding model would include a variance function such as $g(\delta_j) = \sigma_\varepsilon^2 \delta_j^2$, and the syntax would then be

```
R> m3 <- update(m2, var.class = varIdent, weights.form = "method",
+               lme.control = list(opt="optim"))
```

To visualize the summary and graphical output of model m3 we call `summary(m3)` and `lccPlot(m3)`, respectively.

Many other possible models can be built to estimate the LCC through the function `lcc()` options, see Section 1. Model selection can be performed using likelihood-ratio tests for nested models; or using the AIC or BIC criteria, for example,

```
R> AIC(m2, m3); BIC(m2, m3); anova(m2, m3)
```

## EXAMPLES

We will now use three example datasets, drawn from *Lloyd et al. (1998)*, *Martin et al. (2002)* and *Oliveira, Hinde & Zocchi (2018)*, to illustrate the implemented functions in the following sections of this article. The first dataset is an observational study of a cohort of 82

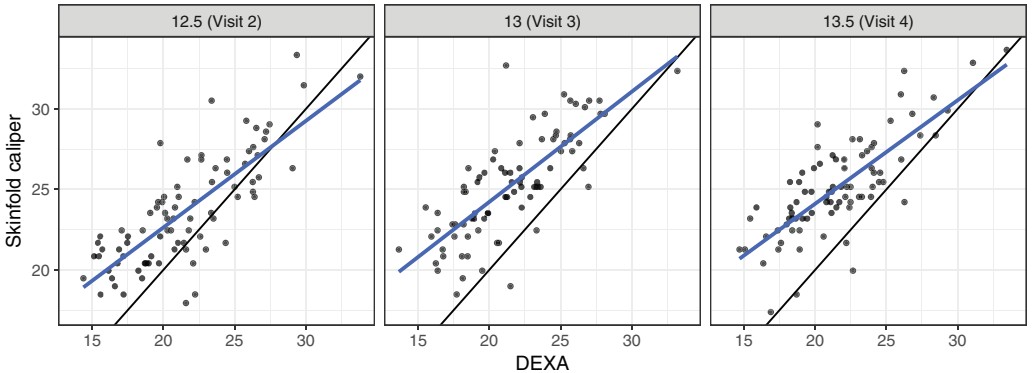

**Figure 1 Scatter plot of body fat data, where the panels represent visits, the blue line is the best fit line, and the black line is the line of equality.**

adolescent females to assess the percentage body fat and the aim is to determine the agreement profile between measurements made over time using a skinfold caliper and dual-energy X-ray absorptiometry. The second is a canonical example from agriculture and was the motivation for the original development of these methods; here the goal is to investigate if a colorimeter can compete with a digital scanner in measuring the peel hue of papayas over time. The final example is again related to medicine and the goal here is to verify the agreement between cortisol concentration measured on patients every hour and every 2 h.

## Percentage body fat dataset

These data came from a longitudinal observational study conducted as part of the Penn State Young Women's Health Study (*Lloyd et al., 1998*). Percentage body fat was measured using skinfold calipers and dual-energy X-ray absorptiometry (DEXA) on a cohort of 82 adolescent white females attending public schools in Pennsylvania. The initial visit occurred at age 12 (baseline) and subsequent visits occurred every 6 months, in which one skinfold caliper and one DEXA measurement were taken to assess the percentage of body fat. As the skinfold measurement is the most frequently used method for laboratory and field studies, the objective was to determine the agreement profile between the skinfold caliper and DEXA measurements. Figure 1 shows that the agreement between skinfold and DEXA apparently decreases over the visits. *King et al. (2007)* explained that this phenomenon may occur because the skinfold method is only capable of detecting subcutaneous fat, while DEXA detects subcutaneous, breast, lower body and visceral fat. Moreover, female adolescents may have a considerable fat increase in breast, lower body and/or visceral fat over this age range (*King et al., 2007*). Consequently, this reinforces the interest in estimating the agreement profile between these methods for the body fat measurements over ages ranging from 12.5 to 13.5 years old, rather than summarizing it in a single coefficient as proposed by *King et al. (2007)*. Hence, we created a new variable called TIME given by $12 \times (age - 12)$, which represents the time in months after the first visit (baseline).

Now let $y_{ijk}$ be the measurement taken on the $i$-th individual, by the $j$-th method at the $k$-th visit. We then fit a random intercepts and slopes linear regression model, given by

$$y_{ijk} = \beta_{0j} + b_{0i} + \left(\beta_{1j} + b_{1i}\right)t_k + \varepsilon_{ijk}$$
$$\boldsymbol{b} = [b_{0i}, b_{1i}]^T \sim N_2(\boldsymbol{0}, \boldsymbol{G}) \quad \text{and} \quad \varepsilon_{ijk} \sim N\left(0, \sigma_\varepsilon^2\right),$$

(4)

where $\text{vech}(\boldsymbol{G}) = \left[\sigma_{b_0}^2, \sigma_{b_{01}}, \sigma_{b_1}^2\right]^T$ (vech($\cdot$) is the half-vectorization of a symmetric matrix $\boldsymbol{G}$ formed from only the lower triangular part). Using model (4), we estimate the LCC, LPC and LA statistics as well as their 95% bootstrap confidence intervals based on 10,000 pseudo-samples using the `lcc()` function:

```
R> data(bfat, package = "cccrm")
R> library(dplyr)
R> bfat <- bfat %>%
+        mutate(VISITNO = replace(VISITNO, VISITNO == 2, 12.5)) %>%
+        mutate(VISITNO = replace(VISITNO, VISITNO == 3, 13)) %>%
+        mutate(VISITNO = replace(VISITNO, VISITNO == 4, 13.5)) %>%
+        mutate(SUBJECT = factor(SUBJECT)) %>%
+        mutate(MET = factor(MET, labels = c("1 hour", "2 hours")))
R> bfat$TIME <- 12 * (bfat$VISITNO - 12)
R> set.seed(134)
R> m.bfat.1 <- lcc(data = bfat, subject = "SUBJECT", resp = "BF",
+        method = "MET", time = "TIME", qf = 1, qr = 1,
+        components = TRUE, ci = TRUE, nboot = 10000)
Convergence error in 902 out of 10,000 bootstrap samples.
```

The output of model `m.bfat.1` indicates that in 902 (9.02%) of the pseudo-samples, the likelihood maximization algorithm failed to converge, where most of these failures were a consequence of specific bootstrap sample patterns. An alternative procedure to decrease the percentage of convergence failures is by increasing the iteration limit and/or changing the optimization method from `nlminb` to `optim`. In the `lcc()` function, the user can include a list of optimisation control additional arguments in the `lme.control()` function:

```
R> set.seed(134)
R> m.bfat.2 <- update(m.bfat.1, lme.control = list(opt = "optim"))
Convergence error in 76 out of 10,000 bootstrap samples.
```

The output of `m.bfat.2` shows a lower number of failures (0.76%) compared with the previous approach. We proceed to examine the bootstrap confidence intervals computed for the LCC, LPC and LA:

```
R> summary(m.bfat.2, type = "lcc")
Longitudinal concordance correlation model fit by REML
AIC          BIC          logLik
2182.068     2215.59      -1083.034
```

```
gof: 0.9201

Lower and upper bound of % bootstrap confidence
Number of bootstrap samples:

DEXA vs. skinfold
$LCC
    Time      LCC         Lower        Upper
1     6    0.6653516   0.5687779   0.7395459
2    12    0.5589258   0.4516374   0.6442955
3    18    0.4588008   0.3353932   0.5599172
$LPC
    Time      LPC         Lower        Upper
1     6    0.8065578   0.7415331   0.8558988
2    12    0.7826493   0.7092871   0.8378992
3    18    0.7620551   0.6676806   0.8300397
$LA
    Time      LA          Lower        Upper
1     6    0.8249273   0.7431156   0.8898124
2    12    0.7141458   0.6201347   0.7923521
3    18    0.6020573   0.4934167   0.6961643
```

We may then plot the LCC, LPC and LA with their respective confidence intervals by executing

```
R> lccPlot(m.bfat.2)
R> lccPlot(m.bfat.2, type = "lpc")
R> lccPlot(m.bfat.2, type = "la")
```

The estimates of LCC, LPC and LA, their confidence intervals, and figures indicate that the agreement and accuracy profiles between the skinfold caliper and DEXA measurements decrease over time, while the precision profile, represented by LPC, remains constant (Fig. 2). Therefore, a first conclusion is that the agreement profile decreases over time because the accuracy is decreasing.

Moreover, there is a moderate to weak agreement profile, where the greatest LCC estimate was 0.6654 at age 12.5 (95% CI [0.5688–0.7395]) and the smallest LCC estimate was 0.4588 at age 13.5 (95% CI [0.3354–0.5599]). This result reinforces the discussion presented by *King et al. (2007)*, who provided physiological explanations for this phenomenon due to fact that the skinfold method is not capable to detect breast, lower body and visceral fat, which increases over this age range. Clearly, as the skinfold method detects less fat than the other, the accuracy between them tends to decrease since the expected value difference is greater (Fig. 2C). The concordance correlation coefficient between fitted values of the mixed-effects model and observed values is presented as goodness of fit (gof) and was approximately 0.92. This result shows that the model can reproduce the observed values quite well.

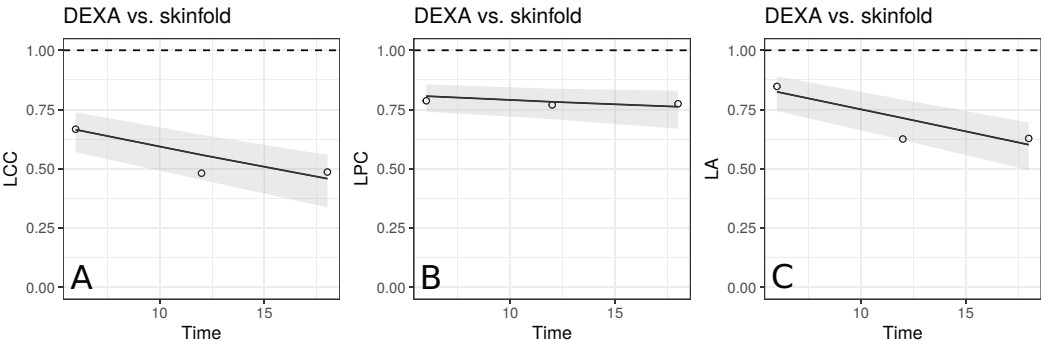

**Figure 2 Estimate and 95% bootstrap confidence interval for the (A) longitudinal concordance correlation (LCC); (B) longitudinal Pearson correlation (LPC); and (C) longitudinal accuracy (LA) between percentage body fat measured on adolescent girls by skinfold caliper and DEXA.** Points represent (A) the sample CCC, (B) sample Pearson correlation and (C) sample accuracy.

## The papaya peel hue dataset

In commercial fruit classification, one of the most important variables is the peel hue because it is used to determine fruit ripeness (*Mendoza & Aguilera, 2004*; *Oliveira, Zocchi & Jacomino, 2017*). This is very important to plan harvesting procedures. In an experiment described in *Oliveira, Hinde & Zocchi (2018)*, the hue component was measured for a sample of 20 papaya fruits using a flat-bed scanner (HP Scanjet G2410) and a colorimeter (Minolta CR-300) (*Konica Minolta, 2003*). The hue of each fruit was measured daily using both devices for a period of 15 days, where four equidistant points on the equatorial region were observed using a colorimeter, and 1,000 points over the same region were observed using a scanner. The circular mean hue was calculated for the $i$th fruit, $i = 1, 2, \ldots, N$, measured by the $j$th method, $j = 1, 2$ at time $t_{ik}$, 390 $k = 1, 2, \ldots,$ $n_i$. As the multivariate von Mises distribution of the hue is highly concentrated around its overall mean, we assume that its distribution can be treated as a normal distribution with mean $\mu_h = 391$ and covariance matrix $\boldsymbol{R} = \boldsymbol{I}\,\sigma_\varepsilon^2$.

The aim of the agreement study here was to determine whether the scanner can reproduce the mean hue measurements taken by the colorimeter on the same fruit over time. The colorimeter is faster and easier to use than a flatbed scanner. Additionally, each image obtained with the scanner needs to processed by an image manipulation program to select the object and extract its pixel-by-pixel information. Our major interest here is in the longitudinal accuracy profile, because high values over time would suggests that the fruit's topography does not influence the measurements taken by the scanner.

We start by making a plot of individual profiles grouped by measurement device, as well as a scatterplot of the hue data (Fig. 3). We fit a second-degree polynomial model over time for each fruit considering all observations taken by both devices, and obtain the 95% confidence intervals for the coefficients (Fig. 3C). Apparently, there is a moderate agreement between the scanner and the colorimeter, which increases as the mean hue decreases. However, this could be due to the smaller number of fruits at the end of the experiment (fruits that presented disease had to be dropped out of the study).

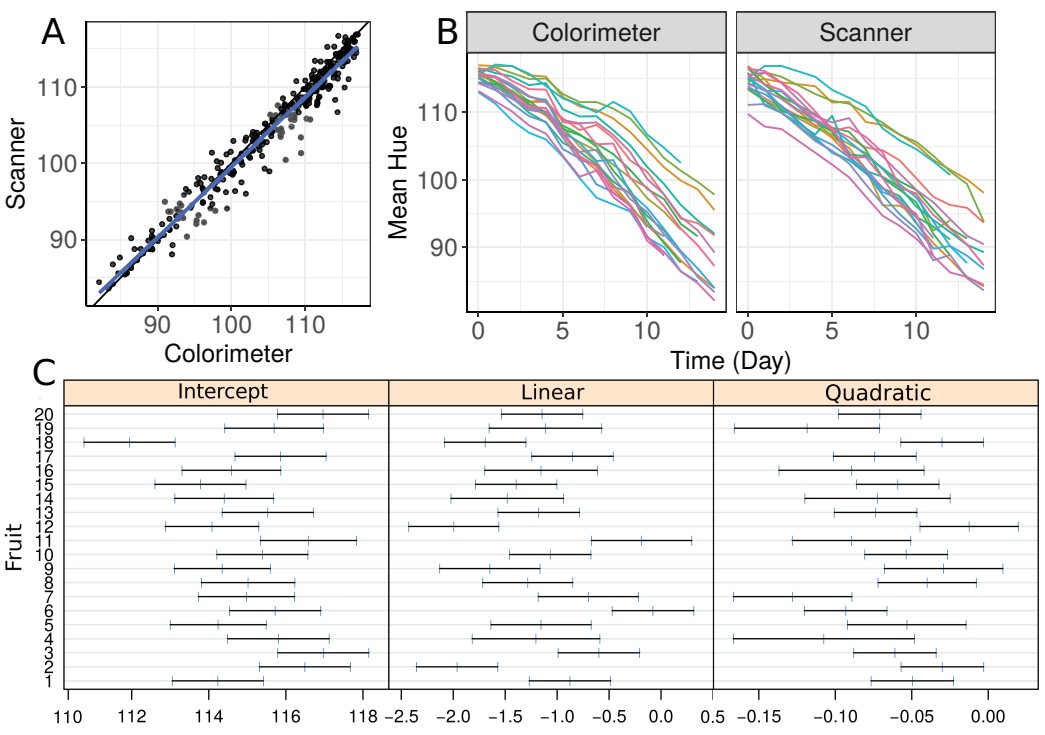

**Figure 3** (A) Scatterplot of hue data considering all repeated measurements with a blue line representing the best fit line and the black one the line of equality, (B) Individual profiles of the peel hue of 20 papaya fruits measured by a colorimeter and a scanner, and (C) individual 95% confidence intervals for second degree polynomial coefficients fitted to the data on each fruit considering all methods together.

Let $y_{ijk}$ be the peel hue measured on fruit $i$, using method $j$ at time point $k$. We start by fitting a second degree polynomial mixed-effects model with random intercepts, linear and quadratic coefficients, written as

$$y_{ijk} = \beta_{0j} + b_{0i} + \left(\beta_{1j} + b_{1i}\right)t_k + \left(\beta_{2j} + b_{2i}\right)t_k^2 + \varepsilon_{ijk},$$

$$\boldsymbol{b} = [b_{0i}, b_{1i}, b_{2i}]^T \sim N_3(\boldsymbol{0}, \boldsymbol{G}) \quad \text{and} \quad \varepsilon_{ijk} \sim N\left(0, \sigma_\varepsilon^2\right),$$

(5)

where $\text{vech}\left(\boldsymbol{G}\right) = \left[\sigma_{b_0}^2, \sigma_{b_{01}}, \sigma_{b_{02}}, \sigma_{b_1}^2, \sigma_{b_{12}}, \sigma_{b_2}^2\right]^T$. Under the model (5), the LCC is given by

$$\rho_{jj'}(t_k) = \frac{\mathbf{t}_k \boldsymbol{G} \mathbf{t}_k^T}{\mathbf{t}_k \boldsymbol{G} \mathbf{t}_k^T + \sigma_\varepsilon^2 + \dfrac{1}{2}S_{jj'}^2(t_k)}.$$

We can fit this model to estimate the LCC, LPC and LA statistics as well as to compute their 95% bootstrap confidence intervals based on 10,000 pseudo-samples using the `lcc()` function directly:

```
R> data(hue)
R> set.seed(6836)
R> m.hue.2 <- lcc(data = hue, subject = "Fruit", resp = "H_mean",
```

```
+               method = "Method", time = "Time", qf = 2, qr = 2,
+               ci = TRUE, nboot = 10000, components = TRUE)
Convergence error in 3133 out of 10000 bootstrap samples.
```

The model used to estimate $\rho_{jj'}(t_k)$ as well as its sampled and fitted values can be extracted by using `summary(m.hue.2, type = "model")` and `summary(m.hue.2, type = "lcc")`, respectively. Moreover, a graphical representation of fitted values and confidence intervals for LCC, LPC and LA can be obtained by executing

```
R> lccPlot(m.hue.2)
R> lccPlot(m.hue.2, type = "lpc")
R> lccPlot(m.hue.2, type = "la")
```

Apparently, the estimated LCC increases over time (Fig. 4A). However, note that it is necessary to check whether the model assumptions were fulfilled because the estimates for the LCC and its bootstrap confidence intervals may be biased under a misspecified model. We therefore checked (i) the normality assumption for the errors, by producing a normal plot of the within-group standardized residuals (Fig. S1A), which indicates that this assumption for the within-group errors is almost plausible, and is not far from a normal distribution; (ii) the homoscedasticity over time was evaluated via a plot of the standardized residuals versus time (Fig. S1B), which indicates an apparent residual correlation for observations taken by the colorimeter and greater between-subject variance for observations taken by the scanner (Fig. S2); (iii) the normality assumption for the random effects (Fig. S1C), which are verified by producing a normal plot for $b_{0i}$, $b_{1i}$ and $b_{2i}$. Additionally, the goodness of fit (`gof`) was 0.992, indicating a high concordance among the model fitted values and observed values. Thus, we update the model `m.hue.2` to include different variances for each level of the factor "method", where the variance function is given by:

$$\text{Var}\left(\varepsilon_{ijk}\right) = \sigma_{\varepsilon}^2 \delta_j^2, \text{ with } j = 1, 2.$$

To ensure identifiability we assume that $\delta_1 = 1$. We also created a regular sequence from the time variable that can be used to make predictions

```
R> lcc_time <- with(hue, list(time = Time, from =min(Time), + to=max
(Time), n=50))
```

This model can be specified in the `lcc()` as

```
R> set.seed(6836)
R> m.hue.3 <- update(m.hue.2, var.class = varIdent,
weights.form = "method",
+               time_lcc = lcc_time,
+               lme.control = lmeControl(opt = "optim"))
Convergence error in 1187 out of 10000 bootstrap samples.
```

As models `m.hue.2` and `m.hue.3` are nested, we can use the likelihood ratio to test the hypothesis $H_0 : \delta_2^2 = 1$ versus $H_a : \delta_2^2 \neq 1$:

```
R> anova(m.hue.2, m.hue.3)
```

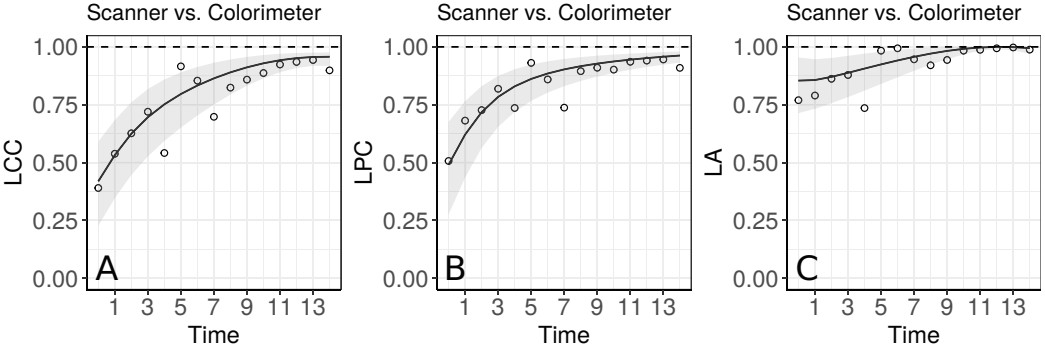

**Figure 4** Estimate and 95% bootstrap confidence interval for the (A) longitudinal concordance correlation (LCC); (B) longitudinal Pearson correlation; and (C) longitudinal accuracy between observations measured by the scanner and the colorimeter with points that represent the (A) sample CCC, (B) sample Pearson correlation coefficient and (C) sample accuracy, using model (5).

```
Model  df     AIC       BIC      logLik    Test  L.Ratio  p-value
1      13  1938.125  1994.107  -956.0625
2      14  1934.920  1995.207  -953.4598  1 vs 2  5.205331  0.0225
```

The result shows that we reject $H_0$ in favour of $H_\alpha$ at a significance level of $\alpha = 0.05$, that is, the inclusion of the function $g(\delta_j) = \delta_j^2$ was significantly important in explaining the extra variability between observations taken at different times.

Moreover, the gof between fitted and observed values for `m.hue.3` model has, practically, the same value as presented for the `m.hue.2` model.

```
R> summary(m.hue.3, type = "lcc")$gof
[1] 0.9915905
```

Although the parameter $\delta_2^2$ was important to explain the variability by method, we can see in Fig. S3 that the model assumptions were still not completely fulfilled because there is a possible correlation among residuals for the colorimeter methodology. However, this model is more plausible than the first one. The sample semivariogram estimate is presented in Fig. S3B and it appears to vary non-randomly around 0.9. Further studies involving the inclusion of correlation structures for the within-group residuals to compute the longitudinal concordance correlation function are still in development.

The agreement profile changes over time, being smaller at the beginning of the experiment and increasing to values close to 1 (Fig. 5). If we consider values above 0.80 for the lower bound of the CI as an indication for interchangeability between the use of the two methods, the colorimeter could be used from the 12th day onwards.

## The blood draw dataset

The blood draw dataset was used as an example in the cccrm package developed by *Carrasco et al. (2013)*. This dataset comes from a study conducted by the Asthma Clinical Research Network (ACRN) (*Martin et al., 2002*). In this double-blinded clinical trial, 144 subjects were randomized to one of six inhaled corticosteroid combinations, and the primary aim of the study was to estimate dose-response curves with respect to adrenal

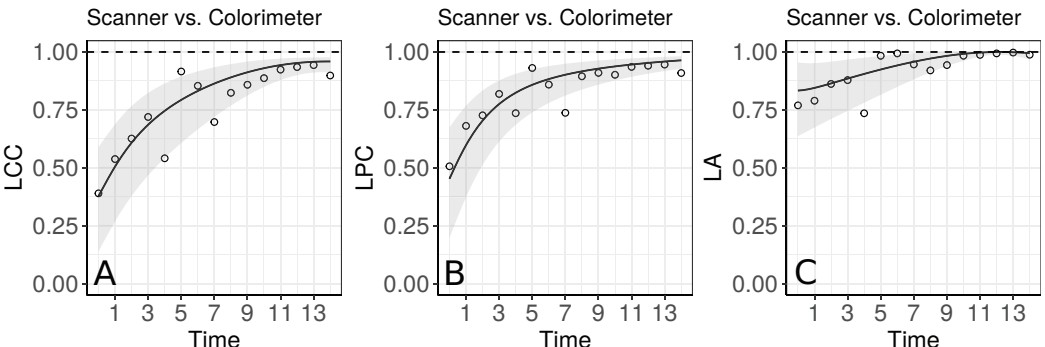

Figure 5 Estimate and 95% bootstrap confidence interval for the (A) longitudinal concordance correlation (LCC); (B) longitudinal Pearson correlation; and (C) longitudinal accuracy between observations measured by the scanner and the colorimeter with points that represent the (A) sample CCC, (B) sample Pearson correlation coefficient and (C) sample accuracy, using the model that estimates different variances for each method.

suppression. After two weeks, the subjects were admitted for overnight testing once a week, for the next five weeks (visits). Blood samples were collected hourly between 8pm and 8am. Then, the plasma cortisol area under the curve (AUC) was calculated using the trapezoidal rule. A secondary objective here was to assess the agreement of the results from blood sampling performed hourly or every two hours, when calculating the plasma cortisol AUC. As an example, we used all individual profiles whose expected value can be described using a second or lower degree polynomial mixed-effects model:

```
R> data(bdaw, package = "cccrm")
R> bdaw$SUBJ <- as.factor(bdaw$SUBJ)
R> bdawMET <- as.factor(bdaw$MET)
R> levels(bdaw$MET) <- c("1 hour", "2 hours")
R> length(unique(bdaw$SUBJ))
R> library(nlme)
R> fit_list <- lmList(AUC ~ poly(VNUM, 4) | SUBJ, data = bdaw)
R> int <- intervals(fit_list)
R> zero_included <- function(x) {
+    flag <- min(x) < 0 & max(x) > 0
+    return(flag)
+  }
R> selected_subj<- names(
+    which(apply(int[,,4], 1, zero_included) &
+    apply(int[,,5], 1, zero_included)))
R> bdaw_subset <- subset(bdaw, SUBJ %in% selected_subj)
```

The scatterplot of the AUC taken every two hours as a function of the AUC taken each hour and plots of the 19 selected individual profiles are presented in Fig. 6.

There seems to be a moderate to strong agreement between the plasma cortisol AUC measurements from blood draw samples taken hourly and every two hours (Fig. 6A). Furthermore, we can also see high variability between subjects and that the AUC decreases

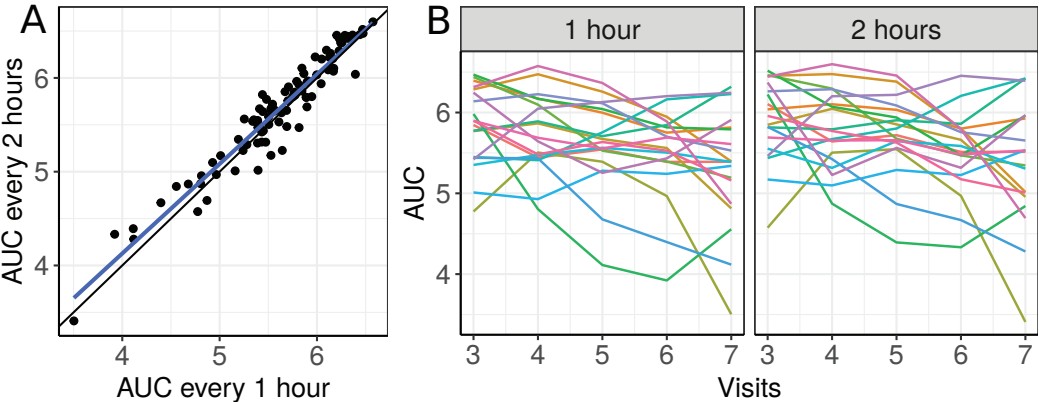

**Figure 6** (A) Scatterplot of the blood draw data considering all repeated measurements (best fit line in blue and equality line in black), and (B) individual profiles of the plasma cortisol AUC calculated from measurements taken every hour and every 2 h.

over time for some subjects (Fig. 6B). We begin by fitting a first degree polynomial model with a subject random intercept and slope model.

```
R> m.bw.1 <- lcc(data = bdaw_subset, subject = "SUBJ",
+              resp = "AUC", method = "MET", time = "VNUM",
+              qf = 1, qr = 1)
R> summary(m.bw.1, type = "lcc")$gof
[1] 0.8850628
```

This model gives only a moderate fit to the data and this is confirmed by the estimated CCC between fitted and sampled values of 0.885 (Fig. 7C). Two possible reasons are (i) we need a higher degree polynomial mixed model to correctly describe some subject profiles, and/or (ii) a possible heteroscedasticity across time, potentially caused by three somewhat different subject profiles, that should be included in the model (Fig. 7B). In addition, the normality assumptions for the within group error and random effects were easily checked by producing the normal plot with simulation envelope (Figs. 7E and 7F) and seem to be broadly plausible.

```
R> plot(m.bw.1, which = c(1, 2, 4, 5, 6))
```

We now fit a second degree polynomial model with random subject effects for all coefficients and compute the 95% bootstrap confidence intervals based on 10.000 bootstrap samples for LCC, LPC and LA components.

```
R> m.bw.2 <- update(m.bw.1, qf = 2, qr = 2, components = TRUE,
+              time_lcc = list(from = 3, to = 7, n = 50),
+              ci = TRUE, nboot = 10000, show.warnings = TRUE,
+              lme.control = lmeControl(msMaxIter = 200,
+              msMaxEval = 600, maxIter = 200), numCore = 4)
Convergence error in 0 out of 10000 bootstrap samples.
```

The summary of the mixed effects model used to estimate LCC, LPC and LA is presented below:

```
R> summary(m.bw.2)
Linear mixed-effects model fit by REML
```

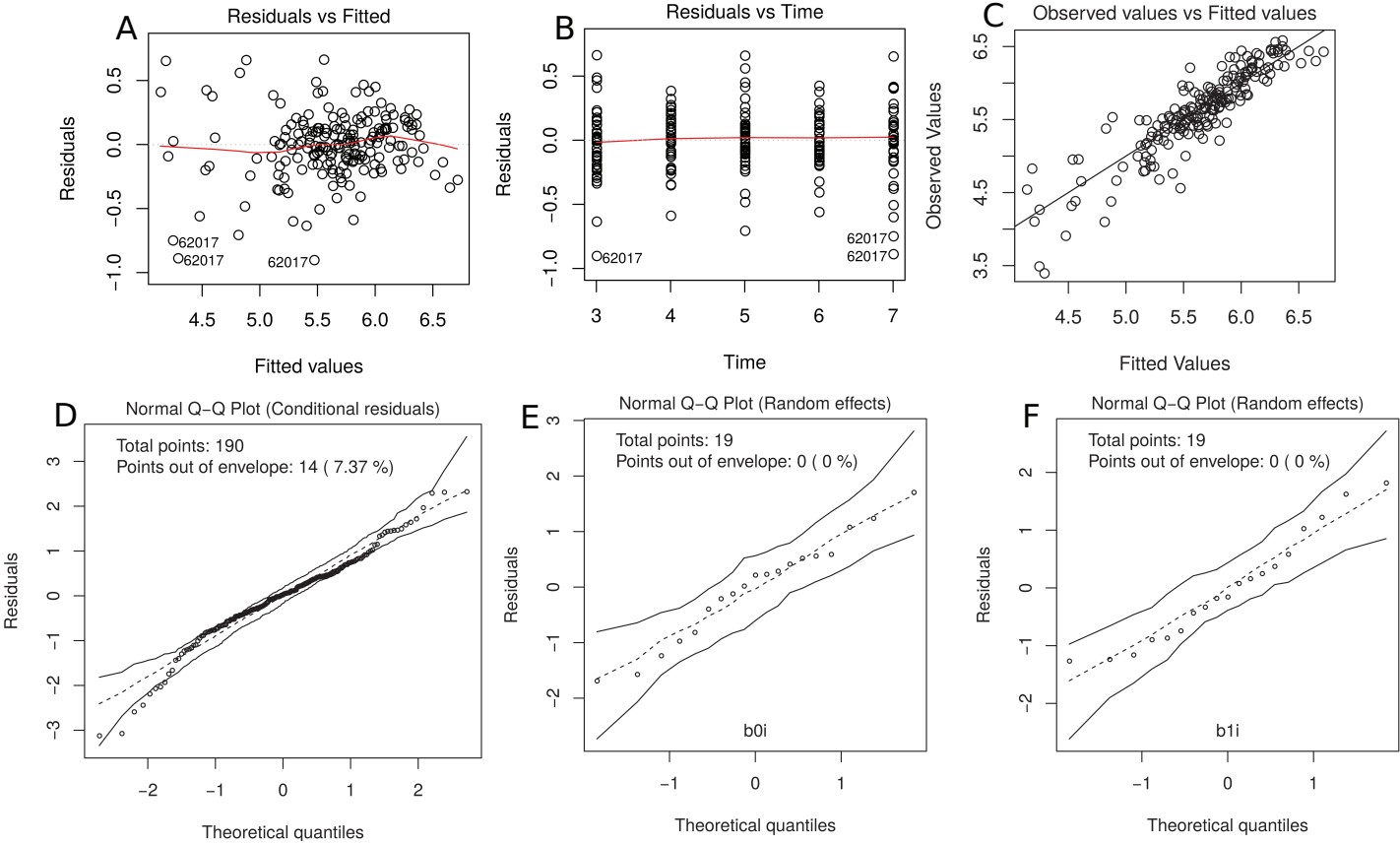

**Figure 7** (A) plot of standardized residuals versus fitted values, (B) standardized residuals versus visits; (C) observed values versus fitted values; (D) normal Q-Q plot with 95% simulation envelop for the conditional residuals; and (E and F) normal Q–Q plot with 95% simulation envelop for random effects.

```
Data: Data
AIC         BIC        logLik
33.93831   75.73247   −3.969153

Random effects:
Formula: ~fmla.rand − 1 | subject
Structure: General positive-definite
                                              StdDev      Corr
fmla.rand(Intercept)                          3.1753653 fm.(I) fd=qr=T
fmla.randpoly(time, degree = qr, raw = TRUE)1 1.3857944 −0.986
fmla.randpoly(time, degree = qr, raw = TRUE)2 0.1404521   0.961 −0.991
Residual                                      0.1269293

Fixed effects: resp ~ fixed − 1
                     Value   Std.Error  DF   t-value  p-value
fixed(Intercept)     6.0147  0.75167    166   8.0018   0.0000
fixedmethod2 hours   0.0471  0.26203    166   0.1796   0.8576
fixedPoly1          −0.0277  0.32744    166  −0.0847   0.9326
```

```
fixedPoly2                    −0.0101  0.03315  166 −0.3046   0.7611
fixedmethod2 hours:Poly1       0.0107  0.11083  166  0.0967   0.9231
fixedmethod2 hours:Poly2      −0.0017  0.01101  166 −0.1500   0.8809
Correlation:
                          fxd(I)   fxdm2h   fxdPl1   fxdPl2   f2h:P1
fixedmethod2 hours        −0.174
fixedPoly1                −0.986    0.167
fixedPoly2                 0.961   −0.160   −0.991
fixedmethod2 hours:Poly1   0.172   −0.989   −0.169    0.165
fixedmethod2 hours:Poly2  −0.168    0.966    0.168   −0.166   −0.993

Standardized Within-Group Residuals:
Min                 Q1            Med            Q3            Max
−2.97645030   −0.48398412   0.03947773   0.59922913   1.87267383

Number of Observations: 190
Number of Groups: 19
```

Now we can test the hypotheses

$$H_0 : \sigma^2_{b_0} > 0, \sigma^2_{b_1} > 0, \sigma_{b_{12}} > 0, \sigma^2_{b_2} = \sigma_{b_{02}} = \sigma_{b_{12}} = 0 \quad \text{vs.} \quad H_a : D \text{ is positive definite}$$

which is equivalent to testing whether the additional variance components of the model m. bw.2 in relation to m.bw.1 are equal to zero:

```
R> m.bw.3 <- update(m.bw.1, qf = 2)
R> anova(m.bw.3, m.bw.2)
         Model  df    AIC      BIC     logLik  Test  L.Ratio  p-value
m.bw.3     1    10   207.642  239.792 −93.821
m.bw.2     2    13    33.938   75.732  −3.969  1 vs 2  179.70   <.0001
```

and these results clearly show that those additional variance components are important. Furthermore, the CCC between fitted and observed values also indicates that model m. bw.2 fits better than model m.bw.1, and m.bw.3.

```
R> summary(m.bw.1, type="lcc")$gof
[1] 0.8850628
R> summary(m.bw.2, type="lcc")$gof
[1] 0.9830078
R> summary(m.bw.3, type="lcc")$gof
[1] 0.8856218
```

Figure 5 shows the fitted LCC, LPC, and LA for concentration of plasma cortisol AUC between measurements taken every hour and taken every 2 h and their respective 95% confidence intervals.

```
R> lccPlot(m.bw.2, control = list(scale_y_continuous = c(0.85, 1)))
R> lccPlot(m.bw.2, type = "lpc",
+          control = list(scale_y_continuous = c(0.85, 1)))
R> lccPlot(m.bw.2, type = "la"
+          control = list(scale_y_continuous = c(0.85, 1)))
```

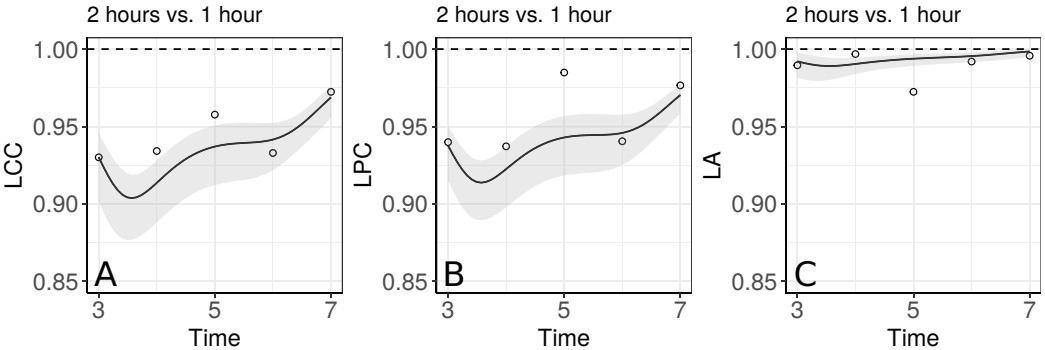

**Figure 8 Estimate and 95% bootstrap confidence interval for (A) longitudinal concordance correlation (LCC); (B) longitudinal Pearson correlation; and (C) longitudinal accuracy for the plasma cortisol AUC between measurements taken every hour and taken every 2 h. In addition, points that represent the sample CCC, sample Pearson correlation coefficient, and sample accuracy, respectively.**

These results show that even though the trend across time is essentially linear at the population level, there is a non-linear trend at the individual level to be more investigated. We can observe that the fitted values and confidence intervals for the LA component were very close to 1 over time, indicating a very high accuracy between methods (Fig. 8C). Consequently, the LCC values depend almost exclusively on the LPC, which indicates a possible problem related to the precision between methods over time, suggesting the use of blood sampled every hour, rather than every two hours, is desirable for this group of patients. It is worthy to note that, as the diagnostic seems broadly plausible for the second degree mixed effects polynomial model (`m.bw.2`), under this model the LCC, LPC, and LA are fourth degree polynomials functions of the time variable.

Additionally, as the `lcc()` function includes the interaction between time and method as default through the argument `interaction = TRUE`, we can test if the interaction effect is necessary using, for example, the following code:

```
R> m.bw.4 <- lcc(data = bdaw_subset, subject = "SUBJ",
+                resp = "AUC",method = "MET", time = "VNUM",
+                qf = 2, qr = 2, REML = FALSE, interaction = FALSE)
R> m.bw.5 <- update(m.bw.4, interaction = TRUE)
R> anova(m.bw.4, fit.bw5)
        Model  df    AIC      BIC     logLik  Test   L.Ratio   p-value
m.bw.4  1      11   -2.5416   33.176  12.271
m.bw.5  2      13    1.2332   43.445  12.383  1 vs 2  0.22520   0.8935
```

The large *p*-value (0.8935) obtained from the likelihood ratio test, as well as the lower AIC and BIC values obtained for model (4) when compared to model (5), suggests no evidence that a model with different slopes describes the data significantly better. Therefore, we opt for the reduced model (4) to analyse the blood draw data. Thus, all of these examples show that our methodology is very flexible and can be applied to many different data types, but the user should be careful about avoiding overfitting. We have also created a Shiny app (https://prof-thiagooliveira.shinyapps.io/lccApp/) using simulated

data in order to stimulate people to learn more about the LCC and verify how each parameter's value can affect the estimation of the LCC, LPC, and LA.

## DISCUSSION

The package `lcc` provides a convenient and versatile tool for estimation and inference about the LCC, LPC and LA. The estimation of these three statistics provides a complete evaluation of the agreement between methods over time (*Oliveira, Hinde & Zocchi, 2018*). These statistics are also very appealing for graphical illustration.

The package supports balanced or unbalanced (dropouts) experimental designs or observational studies, multiple methods, inclusion of covariates in the linear predictor to control systematic variation in the response, and the inclusion of different variance-covariance structures for random-effects and residuals. Residual diagnostic and goodness of fit can be evaluated easily via the generic function `plot()`, which provides up to six built-in diagnostic plots. Furthermore, the `anova()`, `AIC()` and/or `BIC()` functions can be used to aid in model selection.

Statistical inference for the estimators of $\rho_{jj'}(t_k)$, $\rho_{jj'}^{(p)}(t_k)$ and $C_{jj'}(t_k)$ can be obtained using bootstrap confidence intervals based on approximations of their empirical distributions by the normal distribution, or from percentiles of their bootstrap sampling distribution. These methods are, however, computationally intensive.

To the best of our knowledge, there is no package available to estimate the extent of longitudinal agreement between methods. The `lcc` package can be viewed as an extension of the R and SAS `cccrm` package developed by *Carrasco et al. (2013)*. This package handles the time as a factor in the model, and computes the concordance correlation coefficient, which can be viewed as a measure that summarises the interchangeability between methods in relation to all their measurements.

The importance in estimating the LPC, as a measure of precision, and the LA, as a measure of accuracy, was demonstrated in Section 4 (Fig. 5). In particular, both of these statistics can be used jointly to determine if a moderate or small agreement between methods at time $t_k = t$ is related to a precision or an accuracy problem, as suggested by *Lin (1989)*; *Barnhart & Williamson (2001)*; *Lin (1992)*; *Ma et al. (2010)*. In the papaya hue example, the moderate LCC is highly influenced by a moderate LPC, suggesting that if we increase the number of points observed with the colorimeter on the equatorial region up to day 10, the colorimeter will probably be able to reproduce the measurements taken by the scanner. Future studies involve the determination of the sample size over time based on the least acceptable LCC, assuming we can accept up to a certain amount of loss in the LPC and in the LA, as discussed by *Lin (1992)*.

It would be useful to mention that, as a naive analysis, the Bland-Altman method (*Bland & Altman, 1986*) is commonly used to calculate the mean difference between two methods (as a measurement of "bias") with the addition of 95% limits of agreement (LoA) in the analysis of repeated-measures studies (including longitudinal data). If these methods are being compared without a 'golden standard' reference (*Lin, 1989*), an improved Bland-Altman interval approach is preferred (*Liao & Capen, 2011*).

Although these approaches are not suitable to analyse repeated-measures designs, researchers still use it to explore the data because the method is simple to use. However, even if the Bland-Altman method has observations outside of LoA range, two methods can have a very high concordance correlation when the correct variance-covariance structure is accounted for in the model, as discussed by *Zhao et al. (2009)*. This demonstrates the value of the availability of packages that enable the selection of matrix structures for random effects and error term when calculating the longitudinal concordance correlation.

Another interesting remark is that when the systematic difference between methods is zero, the CCC calculated based on a mixed-effects model is equivalent to the intraclass correlation coefficient (ICC) (*Carrasco, King & Chinchilli, 2009*). In the same direction, the ICC as a function of the time variable is a particular case of the longitudinal concordance correlation function when $S_{jj'}^2(t_k) = 0$. If we consider the repeated measures, the ICC gives us the percentage of total variability explained by subject over time, and, consequently, it is not comparable with the LCC in terms of a longitudinal agreement index between methods.

Finally, all examples discussed in Section 4 show that our methodology is flexible, and can be applied to many different data types. One limitation of the lcc package is that, for the time being, the covar argument only allows for including fixed-effect covariates in the linear predictor. We plan to update our package in the near future to handle with the inclusion of fixed-effects and random-effects covariates, as well as interaction effects.

## CONCLUSION

The `lcc` package implements methods to estimate the LCC, LPC and LA functions as well as their bootstrap confidence intervals. In this package, we included different structures for the variance-covariance matrices of random-effects and residuals, allowing estimation of the extent of longitudinal agreement between methods under different assumptions. Functions `plot()`, for diagnostics, `summary()` and `lccPlot()`, for numerical and graphical summaries, respectively, and `anova()`, `AIC()`, `BIC()`, for model selection, make the package flexible and easy to use. Furthermore, the mixed-effects model based approach to compute the LCC allows us to work with both balanced and unbalanced experimental designs and observational studies.

## ACKNOWLEDGEMENTS

This manuscript benefited from insightful comments and suggestions provided by two anonymous reviewers and an academic editor.

### Funding

This work was supported by the CNPq (National Council of Technological and Scientific Development), CAPES (Coordination for the Improvement of Higher Education Personnel), and Science Foundation Ireland (SFI) under grant number SFI/12/RC/2289,

co-funded by the European Regional Development Fund. The funders had no role in study design, data collection and analysis, decision to publish, or preparation of the manuscript.

### Grant Disclosures
The following grant information was disclosed by the authors:
National Council of Technological and Scientific Development (CNPq).
Coordination for the Improvement of Higher Education Personnel (CAPES).
Science Foundation Ireland (SFI): SFI/12/RC/2289.
European Regional Development Fund.

### Competing Interests
The authors declare that they have no competing interests.

### Author Contributions
- Thiago P. Oliveira conceived and designed the experiments, performed the experiments, analyzed the data, prepared figures and/or tables, authored or reviewed drafts of the paper, and approved the final draft.
- Rafael A. Moral analyzed the data, prepared figures and/or tables, authored or reviewed drafts of the paper, and approved the final draft.
- Silvio S. Zocchi conceived and designed the experiments, performed the experiments, prepared figures and/or tables, and approved the final draft.
- Clarice G.B. Demetrio analyzed the data, prepared figures and/or tables, and approved the final draft.
- John Hinde analyzed the data, authored or reviewed drafts of the paper, and approved the final draft.

### Data Availability
   The data used to illustrate the lcc package are publicly available:
   - Body fat data: https://www.rdocumentation.org/packages/cccrm/versions/1.2.1/topics/bfat.
   - Hue data: https://data.mendeley.com/datasets/w2phnmb56j/1.
   - Blood draw data: https://www.rdocumentation.org/packages/cccrm/versions/1.2.1/topics/bdaw.
   - The package lcc is available at the CRAN repository: https://cran.r-project.org/web/packages/lcc/index.html.
   - GitHub: https://github.com/Prof-ThiagoOliveira/lcc.
   All code used to analyse the database, including models, plots, and packages are available in the Supplemental Files.

### Supplemental Information
Supplemental information for this article can be found online at http://dx.doi.org/10.7717/peerj.9850#supplemental-information.

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
