# Peer review of "lcc: an R package to estimate the concordance correlation, Pearson correlation and accuracy over time"

_PeerJ, doi:10.7717/peerj.9850_

## Round 0.1 · original submission · Minor Revisions

As you can see from the attached reports, the reviewers have some minor points to address and gave very constructive feedback. I hope the comments are helpful for you to further improve your manuscript.

Reviewer 1 ·

Basic reporting

I was able to reproduce nearly all the results in the paper, but the three bootstrap results slightly differed. This is easily addressed using set.seed(). In addition, I have suggestions for making reproducibility easier. The figures are very nice.

Major:
a) Bootstrapping Reproducibility: set.seed(random.number) to make bootstrapping return identical results every time. Sometimes I got no convergence errors and other times I got a different number of errors than reported in the paper. Fortunately, this a really easy to address.

set.seed(532)
m.bfat.1 <- lcc(data = bfat, subject = "SUBJECT", resp = "BF", method = "MET", time = "TIME", qf = 1, qr = 1, components = TRUE, ci = T, nboot = 10000)
m.hue.2 ….
m.bw.2 …

Minor:
a) Suggest creating Rmd: It'll be much easier to reproduce the results using an R Markdown file that is self-contained.

b) Suggest loading all required packages and datasets at the start:
if (!require("pacman")) install.packages("pacman")
pacman::p_load(lcc, cccrm, dplyr, reshape2, tidyverse)

#Datasets:
load(file = "simulated_hue_block.RData")
load(file = "simulated_hue.RData")
data(bfat, package = "cccrm")
data(hue)
data(bdaw, package = "cccrm")

Ideally, the two hue datasets could be added to lcc package instead of being loaded separately. This is a minor issue, but it is an extra step to download the data and then load it.

Also, I recommend adding sessionInfo() and saving all the plots in an Rmd.

Experimental design

The research question for the proposed methods is very well defined. I was able to reproduce the results, with one exception noted above with bootstrapping.

Validity of the findings

Overall, the findings are valid and rigorious. Howver, this is one interpretation that I think should be restated.

On line 632: “As the p-value was 0.8935, we can conclude that there is no interaction effect and, consequently, the fitted curves for each level of method over time can be considered parallel.” Non-significance does not necessarily imply the null (no difference), see Greenland et al. (2016), point 6: https://link.springer.com/article/10.1007/s10654-016-0149-3

It would be safer to interpret this result as model 5 and model 4 are not clearly distinguishable using the likelihood test ratio, see Burnham and Anderson’s (2007) book Model Selection and Multimodel Inference. However, the lower AIC and BIC values for model 4 indicate it is more plausible. I think this is a great opportunity to show an example of a more cautious interpretation. It’s not uncommon to see cherry picking of fit indices. Admittedly, it’s messy when fit statistics conflict because the interpretation is no longer straightforward. I would argue that messiness reflects the reality of the data and models.

Additional comments

The paper is very well written and the R package is sound. Documentation of the R package is excellent. I was able to reproduce all the results in the paper, notwithstanding the main issue raised in the basic reporting section; but that is an easy fix. The lcc package has great functionality with support for common functions such as anova() and plot(). Graphs in the paper are impressive and nicely illustrate the models and model diagnositics. I'll definitely be using this package, I have data with mulitple raters for items over time.

Major Comments:

a) Limits of Agreement: I'm going to caveat this first point by saying I have limited knowledge of the biostatistics literature. Still, I interpret limits of agreement (LoA), the range of absolute agreement or exchangability for two methods, as being different than accuracy along the 45 degree diagonal line because the ground truth value is often unknown. Therefore, it would be useful to mention LoA because two measures/methods can be very highly correlated, yet have a wide range of absolute exchangeability. Here’s an example of different measurements tumor size, near perfect concordance correlations with a broad range of exchangeability: https://pubs.rsna.org/doi/pdf/10.1148/radiol.2522081593

The R MethComp package can handle repeated measures for LoA:
https://cran.r-project.org/web/packages/MethComp/index.html
Repeated measures for LoA is specialized area, most LoA is done with independent observations: https://doi.org/10.1093/bja/aem214

Even though the Bland-Altman method papers are very highly cited (one is in the top 10 of all cited papers in biostats), the distinction between limits of agreement and agreement using correlation is still frequently overlooked. To add to the confusion, LoA is also called method comparison, Bland-Altman analysis/method/plot, and Tukey Mean-Difference Plot.

Original papers:
Bland JM, Altman DG. Statistical methods for assessing agreement between two methods of clinical measurement, Lancet, 1986i(pg. 307-10)
Bland JM, Altman DG. Comparing methods of measurement: why plotting difference against standard method is misleading, Lancet, 1995, vol 346 (pg. 1085-7)
Bland JM, Altman DG. Measuring agreement in method comparison studies, Statisical Methods in Medical Research, 1999, vol 8 (pg. 135-60

Good tutorial paper: https://www.ncbi.nlm.nih.gov/pmc/articles/PMC4470095/

b) Incorrect models for time-series data: To demonstrate what not do and also show why Icc is valuable, consider adding examples of incorrect models.

Specifically, treating non-independent data as independent by fitting conventional ICC models. I’m not aware of any examples of this for reliability stats, but in general this is done more often than one would think: see Aarts (2014).
Aarts, E., Verhage, M., Veenvliet, J. V., Dolan, C. V., & Van Der Sluis, S. (2014). A solution to dependency: using multilevel analysis to accommodate nested data. Nature neuroscience, 17(4).

Another incorrect possibility is to average data and then fit a ICC conventional model.

Minor
a) Connections to multilevel modeling: Is the longitudinal correlation ever the same as the intraclass correlation coefficient (ICCs) in multilevel modeling? I think this is only for a particular multilevel model (unconditional growth model)? I remember Singer and Willett’s (2003) book, Applied Longitudinal Data Analysis, mentioned this. Unfortunately, that book is in my office right now.

b) Performance: This is a wish-list item for the future, in the package rather than the manuscript- if possible, any performance improvements would be wonderful. Example run times on an i7-4770 running Windows 10 with BLAS:

- Approximate run time: Hours, probably? I ran the bootstrapping code overnight.
m.bfat.1 <- lcc(data = bfat, subject = "SUBJECT", resp = "BF",
method = "MET", time = "TIME", qf = 1, qr = 1,
components = TRUE, ci = T, nboot = 10000)

- Approximate run time: 9 minutes
m.bfat.2 <- update(m.bfat.1, lme.control = list(opt = "optim"))

I’m guessing this is going to be extremely difficult, especially with multilevel models and because it looks like the Icc uses nlme? I'm pretty sure nlme doesn't have multi-CPU support. lme4 has multi-CPU support for models and bootstrapping. Bootstrapping may be the best starting point for parallelization.

Reviewer 2 ·

Basic reporting

See my comments to the authors

Experimental design

See my comments to the authors

Validity of the findings

See my comments to the authors

Additional comments

The authors developed a nice r-package for a very important statistical and practical concordance/agreement item, which has a very broad application. It is very important to have computational tool ready for advanced statistical methods. There are many r-packages for assessing the agreement of two measurement methods. A most recent one is named "AgreementInterval" which includes commonly used index approaches such as the CCC and the interval approaches along with graphic tools. The current r-package particularly focuses on the agreement for longitudinal data.

1. The authors used the early proposed concordance correlation coefficient (CCC) and the accuracy index C_{b} (Lin, 1989). However, as pointed out by Liao & Lewis (2000), there are many concerns regarding the metrics. For example, the C_{b} sometimes gives unexplainable results, or totally misleading results. To enhance Lin's CCC, Liao (2003) developed a new concordance correlation coefficient built on Lin's CCC by using two random paired measurements to the line of identity and improved the inferential ability of the new method. This approach increased the assessment accuracy. These facts should be mentioned in the introduction section so that the readers/practitioners can use their subject knowledge to judge the appropriateness of the derived metrics.

2. As the authors pointed out in the article, there are many cases where the agreement is needed for the curved data. The authors studied the agreement for a structured longitudinal data. However, the first paper in the literature for agreement in curved data without any structured assumption was proposed in Liao (2005) using a general non-parametric approach. This information should be mentioned in the introduction section so that the readers/practitioners can use their subject knowledge to judge if their data have the defined longitudinal structure.


• Liao, J.J.Z. and Lewis, J. (2000), “A Note on Concordance Correlation Coefficient”, PDA Journal of Pharmaceutical Science and Technology, 54(1), 23 – 26.
• Liao, J.J.Z. (2003), “An Improved Concordance Correlation Coefficient”. Pharmaceutical Statistics, 2(4), 253 – 261.
• Liao, J.J.Z. (2005), “Agreement for Curved Data”, J. of Biopharmaceutical Statistics, 15, 195 – 203.
• CRAN - Package AgreementInterval:
https://cran.r-project.org/web/packages/AgreementInterval/index.html

---

## Round 0.2 · Minor Revisions

As you can see one reviewer has still a minor comment regarding the Bland-Altman method in the discussion section.

Reviewer 2 ·

Basic reporting

NA

Experimental design

NA

Validity of the findings

NA

Additional comments

The authors answered my early comments. However, a new comment is about the discussion statement for the Bland-Altman method in the discussion section in the updated version. Many discussions can be found about this Bland-Altman method. As mentioned in the review paper by Scholz, et al. (2015), when two methods/techniques are being compared without a ‘golden standard’ or ‘true’ reference which is usually the case in applications, an improved Bland-Altman approach by Liao and Capen (2011) should be the first choice as the statistic procedure. The authors can found out the advantages of the improved Bland-Altman method in detail in Liao and Capen (2011). I do think it is important and necessary to mention this fact there.

Liao, J.J.Z. and Capen, R. (2011), “An Improved Bland-Altman Method for Concordance Assessment”, The International Journal of Biostatistics, Vol. 7: Iss. 1, Article 9, 1 – 17.

Scholz, et. al., 2015, “Non-invasive methods for the determination of body and carcass composition in livestock: Dual-energy X-ray absorptiometry, computed tomography, magnetic resonance imaging and ultrasound: Invited review”, in animal 9(07):1-15

---

## Author Rebuttal · Round 0.2

# Response to reviewers

May 22, 2020

Dear Daniel Fischer,

On behalf of all co-authors, we would like to sincerely thank you for giving us such constructive feedback to improve the impact of our manuscript entitled *lcc: an R package to estimate the concordance correlation, Pearson correlation, and accuracy over time*. We extend our thanks for the valuable time devoted to evaluating and testing our R code. We have revised our text according to all comments from the Editor and both Reviewers, and we believe we now present a much improved manuscript and accompanying software. We provide a point-by-point response to all queries below.

## 1 REVIEWER 1

### 1.1 BASIC REPORTING

1. I was able to reproduce nearly all the results in the paper, but the three bootstrap results slightly differed. This is easily addressed using `set.seed()`. In addition, I have suggestions for making reproducibility easier. The figures are very nice.

   Bootstrapping Reproducibility: set.seed(random.number) to make bootstrapping return identical results every time. Sometimes I got no convergence errors and other times I got a different number of errors than reported in the paper. Fortunately, this a really easy to address.

   ```
   set.seed(532)
   m.bfat.1 <- lcc(data = bfat, subject = "SUBJECT", resp = "BF",
   +               method = "MET", time = "TIME", qf = 1, qr = 1,
   +               components = TRUE, ci = T, nboot = 10000)
   ```

- We agree with the Reviewer that reproducibility is a fundamental requirement to obtain consistent results using the same input data and code. We included a seed (using `set.seed()`) in the code to make it fully reproducible. All results obtained were updated on the manuscript.

2. Suggest creating Rmd: It'll be much easier to reproduce the results using an R Markdown file that is self-contained.

   - We thank the Reviewer for this valuable suggestion, and have created an R Markdown file, making the results easy to reproduce.

3. Suggest loading all required packages and datasets at the start:

```
if (!require("pacman")) install.packages("pacman")
pacman::p_load(lcc, cccrm, dplyr, reshape2, tidyverse)
#Datasets:
load(file = "simulated_hue_block.RData")
load(file = "simulated_hue.RData")
data(bfat, package = "cccrm")
data(hue)
data(bdaw, package = "cccrm")
```

   - We agree with the Reviewer in loading all data sets and packages at beginning of R code. We are thankful to them for sharing an easy way to check, install, and load packages using the `pacman` package.

4. Ideally, the two hue datasets could be added to lcc package instead of being loaded separately. This is a minor issue, but it is an extra step to download the data and then load it.

   - Actually, both datasets are available in the `lcc` package and can be loaded through the `data()` function. In consideration of the Reviewer's comment, we included in the manuscript R code for loading both datasets using that generic function.

5. Also, I recommend adding sessionInfo() and saving all the plots in an Rmd.

   - We followed the recommendation of adding an Rmd file with `sessionInfo()` as Supplementary Material. In addition, we saved all models as an RData to save time when compiling the Rmd, should the user choose to do this.

## 1.2 EXPERIMENTAL DESIGN

1. The research question for the proposed methods is very well defined. I was able to reproduce the results, with one exception noted above with bootstrapping.

   - Thank you for the very pertinent comment. We have included `set.seed()` before computing each bootstrap sample, and the code is now fully reproducible.

## 1.3 VALIDITY OF THE FINDINGS

1. Overall, the findings are valid and rigorious. However, this is one interpretation that I think should be restated.

   On line 632: "As the p-value was 0.8935, we can conclude that there is no interaction effect and, consequently, the fitted curves for each level of method over time can be considered parallel". Non-significance does not necessarily imply the null (no difference), see Greenland et al. (2016), point 6: `https://link.springer.com/article/10.1007/s10654-016-0149-3`.

   It would be safer to interpret this result as model 5 and model 4 are not clearly distinguishable using the likelihood test ratio, see Burnham and Anderson's (2007) book Model Selection and Multimodel Inference. However, the lower AIC and BIC values for model 4 indicate it is more plausible. I think this is a great opportunity to show an example of a more cautious interpretation. It's not uncommon to see cherry picking of fit indices. Admittedly, it's messy when fit statistics conflict because the interpretation is no longer straightforward. I would argue that messiness reflects the reality of the data and models.

   - *We agree with the Reviewer and we have reworded this section in order to reflect a more careful interpretation. The new text reads:*

     *The large p-value (0.8935) obtained from likelihood ratio test, as well as lower AIC and BIC values obtained for model (4) when compared to model (5), suggests no evidence that a model with different slopes describes the data significantly better. Therefore, we opt for the reduced model (4) to analyse the blood draw data.*

## 1.4 COMMENTS FOR THE AUTHOR

1. The paper is very well written and the R package is sound. Documentation of the R package is excellent. I was able to reproduce all the results in the paper, notwithstanding the main issue raised in the basic reporting section; but that is an easy fix. The lcc package has great functionality with support for common functions such as anova() and plot(). Graphs in the paper are impressive and nicely illustrate the models and model diagnositics. I'll definitely be using this package, I have data with multiple raters for items over time.

   - *We thank the Reviewer for the general appraisal, and are happy to know that they will be using our package.*

2. Limits of Agreement: I'm going to caveat this first point by saying I have limited knowledge of the biostatistics literature. Still, I interpret limits of agreement (LoA), the range of absolute agreement or exchangability for two methods, as being different than accuracy along the 45 degree diagonal line because the ground truth value is often unknown.

   Therefore, it would be useful to mention LoA because two measures/methods can be very highly correlated, yet have a wide range of absolute exchangeability. Here's an example of different measurements tumor size, near perfect concordance correlations

with a broad range of exchangeability: `https://pubs.rsna.org/doi/pdf/10.1148/radiol.2522081593`

The R MethComp package can handle repeated measures for LoA: `https://cran.r-project.org/web/packages/MethComp/index.html` Repeated measures for LoA is specialized area, most LoA is done with independent observations: `https://doi.org/10.1093/bja/aem214`

Even though the Bland-Altman method papers are very highly cited (one is in the top 10 of all cited papers in biostats), the distinction between limits of agreement and agreement using correlation is still frequently overlooked. To add to the confusion, LoA is also called method comparison, Bland-Altman analysis/method/plot, and Tukey Mean-Difference Plot.

Original papers: Bland JM, Altman DG. Statistical methods for assessing agreement between two methods of clinical measurement, Lancet, 1986i(pg. 307-10) Bland JM, Altman DG. Comparing methods of measurement: why plotting difference against standard method is misleading, Lancet, 1995, vol 346 (pg. 1085-7) Bland JM, Altman DG. Measuring agreement in method comparison studies, Statisical Methods in Medical Research, 1999, vol 8 (pg. 135-60

Good tutorial paper: `https://www.ncbi.nlm.nih.gov/pmc/articles/PMC4470095/`

- We agree with the Reviewer's interpretation on Limits of Agreement (LoA). LoA combine the mean of differences ($d$) between methods and the standard deviation (s) $d \pm 1.96s$, which allows for the identification of outliers as well as the examination of the trend through e.g. linear regression models.

  When we use $C_b$, the accuracy along the 45-degree identity line, if one method provides the "true values", the absence of a systematic difference implies that there is no bias, and then the observations will be around the identity line, and, consequently, the variability will be controlled by the Pearson correlation only. However, a non-significant result for the systematic differences indicates only that there is no evidence of a systematic effect, even if such systematic effect may exist.

  We have incorporated this in the discussion section of the manuscript. The added text reads:

  *It would be useful to mention that, as a naive analysis, the Bland-Altman method (Bland & Altman, 1986) is commonly used to calculate the mean difference between two methods (as a measurement of "bias") with the addition of 95% limits of agreement in the analysis of repeated-measures studies (including longitudinal data). Although this approach is not suitable to analyse repeated-measures designs, researchers still use it to explore the data because the method is simple to use. However, even if the Bland-Altman method has a wide range of absolute exchangeability, two methods can have a very high concordance correlation when the correct variance-covariance structure is accounted for in the model, as discussed by Zhao et al (2009). This demonstrates the value of the availability of packages that enable the selec-*

*tion of matrix structures for random effects and error term when calculating the longitudinal concordance correlation.*

3. Incorrect models for time-series data: To demonstrate what not do and also show why Icc is valuable, consider adding examples of incorrect models.

   Specifically, treating non-independent data as independent by fitting conventional ICC models. I'm not aware of any examples of this for reliability stats, but in general this is done more often than one would think: see Aarts (2014).

   Aarts, E., Verhage, M., Veenvliet, J. V., Dolan, C. V., & Van Der Sluis, S. (2014). A solution to dependency: using multilevel analysis to accommodate nested data. Nature neuroscience, 17(4).

   Another incorrect possibility is to average data and then fit a ICC conventional model.

   Connections to multilevel modeling: Is the longitudinal correlation ever the same as the intraclass correlation coefficient (ICCs) in multilevel modeling? I think this is only for a particular multilevel model (unconditional growth model)? I remember Singer and Willett's (2003) book, Applied Longitudinal Data Analysis, mentioned this. Unfortunately, that book is in my office right now.

   - The ICC is not necessarily the same as the concordance correlation coefficient based on repeated measurements (CCCrm). Carrasco et al. (2009) proved that CCCrm calculated based on a multilevel model can be equivalent to an extension of the ICC when a matrix of weights $D$ (as proposed by King et al., 2007), can be considered as a diagonal matrix. In this sense, the main difference between the ICC and the CCC is that the latter also accounts for an extra variability the authors called a "systematic difference between observers-time variability". In essence, it is the squared difference between any pair of method means at time $t_k$.

     In this sense, when the systematic difference between methods is zero, i.e.

     $$E[Y_{method1}] - E[Y_{method2}] = 0,$$

     the CCC based on a mixed-effects model is equivalent to the ICC. In the same direction, the ICC as a function of the time variable is a particular case of the longitudinal concordance correlation function when $S^2_{jj'}(t_k) = 0$. Further information about $S^2_{jj'}(t_k)$ can be found in Oliveira et al. (2018).

     Thus, if we consider the repeated measures, the ICC gives us the percentage of total variability explained by Subject over time, and, consequently, it is not comparable with the LCC in terms of a longitudinal agreement index between methods.

     This has been included in the discussion of the paper.

     References:

     Carrasco, JL; King, TS; Chinchilli, VM. (2007) The concordance correlation coefficient for repeated measures estimated by variance components. **Journal of Biopharmaceutical Statistics**, 19: 90-105.

King, TS; Chinchilli, VM; Wang, KL. (2007) A class of repeated measures concordance correlation coefficients. **Journal of Biopharmaceutical Statistics**, 17: 653-672.

Oliveira, TP; Hinde, J.; Zocchi, SS. (2018) Longitudinal Concordance Correlation Function Based on Variance Components: An Application in Fruit Color Analysis, **Journal of Agricultural, Biological, and Environmental Statistics**, v. 23, no. 2, 233-254. `https://doi.org/10.1007/s13253-018-0321-1`

4. Performance: This is a wish-list item for the future, in the package rather than the manuscript- if possible, any performance improvements would be wonderful. Example run times on an i7-4770 running Windows 10 with BLAS:

   a) Approximate run time: Hours, probably? I ran the bootstrapping code overnight.

   ```
   m.bfat.1 <- lcc(data = bfat, subject = "SUBJECT", resp = "BF",
   +               method = "MET", time = "TIME", qf = 1, qr = 1,
   +               components = TRUE, ci = T, nboot = 10000)
   ```

   b) Approximate run time: 9 minutes

   ```
   m.bfat.2 <- update(m.bfat.1, lme.control = list(opt = "optim"))
   ```

   - Model `m.bfat.1` runs in 18.02 minutes while `m.bfat.2` runs in 12.66 minutes when executed on Dell Inspiron 17 7000 with 10th Generation Intel® Core TM i7 processor, 1.80GHz × 4 processor speed, 16GB random access memory (RAM) plus 20GB of swap space, 64-bit integers, and the platform used is a Linux Mint 19.2 Cinnamon system version 5.2.2-050202-generic. However, since `nlminb` is the default method in `nlme`, we decided to continue to use the same default optimization method.

I'm guessing this is going to be extremely difficult, especially with multilevel models and because it looks like the lcc uses nlme? I'm pretty sure nlme doesn't have multi-CPU support. lme4 has multi-CPU support for models and bootstrapping. Bootstrapping may be the best starting point for parallelization.

- We agree with the Reviewer about `nlme` doesn't allowing for multi-CPU support, however since the `lcc` bootstrapping just depends on `nlme` to fit the model, we were able to extend the bootstrap computation using parallel cores. Even though the model fitting procedure doesn't allow for parallelization, we can now make the bootstrapping procedure parallelized. In the light of their suggestion, we included a new argument called `numCore` in the `lcc` package version 1.1.0 (now available from github (https://github.com/Prof-ThiagoOliveira/lcc), and CRAN), which establishes the number of cores used during bootstrapping (defaults to 1 core). Although `lme4` allows for multi-CPU support and new optimization methods, it doesn't support the variance-covariance matrix classes supported by the

`nlme` package. In future releases, we'll include, as alternative approaches, specific functions to calculate the lcc through `lme4` for specific variance-covariance structures.

## 2 Reviewer 2

### 2.1 Comments for the author

1. The authors developed a nice r-package for a very important statistical and practical concordance/agreement item, which has a very broad application. It is very important to have computational tool ready for advanced statistical methods. There are many r-packages for assessing the agreement of two measurement methods. A most recent one is named "AgreementInterval" which includes commonly used index approaches such as the CCC and the interval approaches along with graphic tools. The current r-package particularly focuses on the agreement for longitudinal data.

   - We thank the Reviewer for the general appraisal.

2. The authors used the early proposed concordance correlation coefficient (CCC) and the accuracy index $C_b$ (Lin, 1989). However, as pointed out by Liao & Lewis (2000), there are many concerns regarding the metrics. For example, the $C_b$ sometimes gives unexplainable results, or totally misleading results. To enhance Lin's CCC, Liao (2003) developed a new concordance correlation coefficient built on Lin's CCC by using two random paired measurements to the line of identity and improved the inferential ability of the new method. This approach increased the assessment accuracy. These facts should be mentioned in the introduction section so that the readers/practitioners can use their subject knowledge to judge the appropriateness of the derived metrics.

   - We thank the Reviewer for this suggestion and have added text to the Introduction reflecting the proposed changes. The new added text reads:

     *In an attempt to improve the inferential ability, Liao (2003) extended the concordance correlation coefficient by using two random paired measurements to the identity line.*

3. As the authors pointed out in the article, there are many cases where the agreement is needed for the curved data. The authors studied the agreement for a structured longitudinal data. However, the first paper in the literature for agreement in curved data without any structured assumption was proposed in Liao (2005) using a general non-parametric approach. This information should be mentioned in the introduction section so that the readers/practitioners can use their subject knowledge to judge if their data have the defined longitudinal structure.

   - We thank the Reviewer for this suggestion and have incorporated it in the Introduction section of the manuscript. The added text reads:

     *Nevertheless, sometimes the researcher is not interested in reducing the CCC for repeated measurements to a single value, as proposed by Carrasco et al. (2009) and*

*Carrasco et al. (2013), but in describing the extent of agreement between methods over time, as discussed by Liao (2005) in a non-parametric case.*

---

## Round 0.3 · accepted · Accept

All comments raised by the reviewers have been addressed properly, so my decision is to accept the manuscript for publication.

---

## Author Rebuttal · Round 0.3

# Response to reviewers

July 17, 2020

Dear Daniel Fischer,

On behalf of all co-authors, we would like to sincerely thank you and the Reviewers for giving us this feedback. We have revised our text according to the comments from Reviewer 2.

## 1 REVIEWER 2

### 1.1 COMMENTS FOR THE AUTHOR

1. Thank you for let us know about the extension of Bland-Altman method in the case where there is no gold assay. We have included an paragraph in the discussion section as recommend by the Reviewer. See the text below:

   "If these methods are being compared without a 'golden standard' reference (Lin, 1989), an improved Bland-Altman interval approach is preferred (Liao and Capen, 2011)."